# SᴀᴛLM: Satisfiability-Aided Language Models Using Declarative Prompting

**Xi Ye**    **Qiaochu Chen**    **Isil Dillig**    **Greg Durrett**
Department of Computer Science
The University of Texas at Austin
{xiye,qchen,isil,gdurrett}@cs.utexas.edu

## Abstract

Prior work has combined chain-of-thought prompting in large language models (LLMs) with programmatic representations to perform effective and transparent reasoning. While such an approach works well for tasks that only require forward reasoning (e.g., straightforward arithmetic), it is less effective for constraint solving problems that require more sophisticated planning and search. In this paper, we propose a new *satisfiability-aided language modeling* (SᴀᴛLM) approach for improving the reasoning capabilities of LLMs. We use an LLM to generate a *declarative task specification* rather than an imperative program and leverage an off-the-shelf automated theorem prover to derive the final answer. This approach has two key advantages. The declarative specification is closer to the problem description than the reasoning steps are, so the LLM can parse it out of the description more accurately. Furthermore, by offloading the actual reasoning task to an automated theorem prover, our approach can guarantee the correctness of the answer with respect to the parsed specification and avoid planning errors in the solving process. We evaluate SᴀᴛLM on 8 different datasets and show that it consistently outperforms program-aided LMs in the imperative paradigm. In particular, SᴀᴛLM outperforms program-aided LMs by 23% on a challenging subset of the GSM arithmetic reasoning dataset; SᴀᴛLM also achieves a new SoTA on LSAT and BᴏᴀʀᴅɢᴀᴍᴇQA, surpassing previous models that are trained on the respective training sets.[1]

## 1   Introduction

Using large language models (LLMs) to perform complex reasoning has been a central thrust of recent research (Brown et al., 2020; Chowdhery et al., 2022; Rae et al., 2021; Zhang et al., 2022b). Techniques like scratchpads (Nye et al., 2021) or chain-of-thought prompting (CoT) (Wei et al., 2022c) enable LLMs to follow a sequence of reasoning steps before making a prediction. This paradigm is effective on various multi-step reasoning tasks, especially those with fixed forward reasoning procedures (Wei et al., 2022c), e.g., concatenating the last letters of several words. However, CoT prompting can fall short when scaling to problems that involve intensive computation (Gao et al., 2023) or long sequences of reasoning steps (Creswell et al., 2023; Saparov and He, 2023; Ribeiro et al., 2023).

Solving a complex reasoning problem involves three conceptual components: parsing a natural language description into a representation of the problem, deriving a plan to solve the problem, and executing that plan to obtain an answer. Recent work on improving CoT prompting focuses on fixing *execution errors* by augmenting LLMs with symbolic executors such as a Python interpreter, which

---

[1]Code available at `https://github.com/xiye17/SAT-LM`.

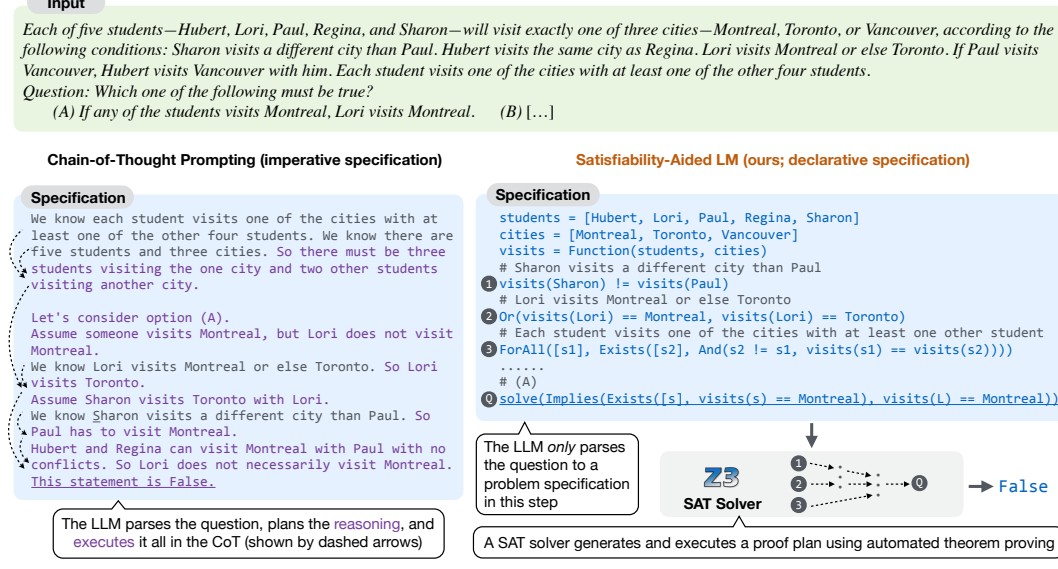

Figure 1: Illustration of our Satisfiability-aided Language Modeling approach (right). We first parse an NL input into a declarative task specification (a set of logical constraints) using prompting (Section 3.1), then use a SAT solver to solve the problem (Section 3.2). The chain-of-thought strategy in prior work (left) yields imperative reasoning processes.

leads to improved performance on arithmetic and symbolic reasoning tasks (Gao et al., 2023; Chen et al., 2022; Lyu et al., 2023). However, CoT prompting (Wei et al., 2022c; Nye et al., 2021) and its executor-augmented successors (Gao et al., 2023; Chen et al., 2022; Lyu et al., 2023) are oriented towards *imperative* solving procedures: a CoT or a program specifies the reasoning procedure as chained steps (Wei et al., 2022c; Gao et al., 2023) in the order of execution. While this is effective for problems whose natural language already provides a suitably clear "plan" for the reasoning, it only leads to limited success for reasoning problems like in Figure 1 that do not outline such a plan (Ribeiro et al., 2023). These problems often state a set of premises and constraints and ask questions that require sophisticated planning to deductively reason over the inputs, which is still challenging even for modern LLMs (Valmeekam et al., 2022).

Our work tackles both execution errors and, more importantly, *planning errors*. We propose SATisfiablity-aided Language Modeling (SATLM) using declarative prompting. The core idea is to cast a natural language (NL) reasoning problem as a satisfiability (SAT for short) problem. As shown in Figure 1 (right), given a problem in NL, we prompt an LLM to parse it into a SAT problem specification which consists of a set of logical formulas, then obtain the solution by invoking a SAT solver.[2] The LLM is specialized towards understanding the preconditions stated in the problem, while the solver is leveraged to plan out the reasoning procedure. In addition, the solver guarantees the correctness of execution, similar to the interpreter used in program-aided LMs (PROGLM).

We evaluate our approach on 8 datasets spanning 4 tasks, including arithmetic reasoning, logical reasoning, symbolic reasoning, and a regex synthesis task. Our SATLM consistently outperforms CoT and PROGLM across all datasets, usually by a large margin. On GSM-SYS, SATLM outperforms PROGLM by a 23%; on GSM, SATLM achieves 84.8% with self-consistency decoding using few-shot prompting, equaling past work that uses the full training set and the same LLM (Li et al., 2022b; Ni et al., 2023). SATLM also sets a new SoTA on LSAT (Zhong et al., 2022), BOARDGAMEQA (Kazemi et al., 2023), and STRUCTUREDREGEX (Ye et al., 2020).

Our analysis illustrates why the combination of SAT solver and declarative prompting is so effective. We find (1) program-aided LMs often make planning errors (e.g., manipulating equations incorrectly), which can be remedied by the SAT solver. (2) Forcing LLMs to explicitly state a declarative

---

[2]Here, we use SAT solver to refer to any automated reasoning tool for checking the satisfiability of formulas in formal logic. Hence, "SAT solver" in this paper also includes first-order theorem provers and SMT solvers.

specification can even improve vanilla CoT prompting. (3) Our SATLM approach can abstain from making uncertain predictions if it parses a problem into an unsatisfiable or ambiguous specification, giving it even higher accuracy in the selective prediction setting (El-Yaniv and Wiener, 2010).

## 2 Overview

This work addresses the challenge of using LLMs to solve NL reasoning tasks. At a high level, an NL reasoning task is a natural language description of a collection of facts $\Phi$ (such as propositions or constraints) about some objects and a question $Q$ related to these objects. The goal of the reasoning task is to find an answer to $Q$ that can be deduced from the information provided in $\Phi$.

We conceptualize the general procedure for solving NL reasoning tasks in three steps: *parsing*, *planning*, and *execution*. We are given natural language input $x_{\texttt{test}} = (NL(\Phi), NL(Q))$ which describes both $\Phi$ and $Q$. Our first step is to parse this natural language into a predicted *task specification* $(\hat{\Phi}, \hat{Q})$, which is a *formal* description of the facts and the query.

Given $(\hat{\Phi}, \hat{Q})$, the planning step then involves determining a sequence of reasoning steps $[r_1, \ldots, r_n]$ beginning with the task specification and ending with the answer to the question. Each step involves invoking a function (e.g., arithmetic operator or logical operator) that produces intermediate results which can be utilized in subsequent steps. A plan can be formulated by an LLM with CoT prompting or by a symbolic solver as in our work here. Finally, we execute the plan systematically with either a symbolic executor (our method) or an LLM, returning the output of the last step, $r_n$, as the answer.

Our solution approaches the problem using exactly these three steps.

**Parsing into declarative specification** We prompt an LLM to generate a specification $s_{\texttt{test}}$ for $x_{\texttt{test}}$. Note that the translation from this description into the specification is not straightforward and cannot be done in a rule-based way for most tasks; Figure 4 shows some particularly complex examples involving commonsense reasoning. The specification $s_{\texttt{test}}$ is a sequence of interleaved NL statements and logical formulas (LF): $s_{\texttt{test}} = [z_1, \ldots, z_n]$ and $z_i \in \Sigma_{NL} \cup \Sigma_{LF}$, where $\Sigma_{NL}$ and $\Sigma_{LF}$ denote the space of natural language and logical formulas, respectively. We derive the formal specification $(\hat{\Phi}, \hat{Q})$ by taking all the $z_i$ in $\Sigma_{LF}$ from $s_{\texttt{test}}$. An example of the specification is presented on the right of Figure 1. Our specification is declarative since we do not explicitly generate the $r_i$ from the LLM at this stage.

**Planning and execution with a SAT solver** Given the predicted formal specification $(\hat{\Phi}, \hat{Q})$, we wish to derive the final answer of the query $\hat{Q}$ from it. We say that a solution $a$ is correct if $\hat{\Phi}$ logically entails $\hat{Q} = a$, denoted as $\hat{\Phi} \models \hat{Q} = a$. The key insight behind our work is to offload *both* the planning and execution steps to a SAT solver. Specifically, we use a SAT solver to find a satisfying assignment for $a$ in the formula:

$$\forall V. (\hat{\Phi} \Rightarrow \hat{Q} = a)$$

where $V$ denotes the set of all variables used in $\hat{\Phi}$ and $\hat{Q} \in V$ is a variable that corresponds to the solution. Note that the only free variable in this formula is $a$; hence, the assignment to $a$ returned by the solver is the final answer to the reasoning problem.

The approach outlined above has two important strengths. First, because the SAT solver is *sound* (i.e., any assignment it produces satisfies the formula), the solution is correct by construction. Thus, assuming that the parsing is correct and $\hat{\Phi}$ and $\hat{Q}$ match $\Phi$ and $Q$, we have a proof that the solution is indeed correct. Second, the planning step is done internally to the solver, and the chain of reasoning steps $[r_1, \ldots, r_n]$ can be obtained by asking the solver to produce a proof of the validity of the formula $\hat{\Phi} \Rightarrow \hat{Q} = a^*$ where $a^*$ is the assignment produced by the SAT solver. All solvers we consider can produce such a proof of validity (e.g., in the form of a resolution refutation (Davis and Putnam, 1960)).

**Comparison with prior work** Prior approaches to NL-based reasoning with LLMs can also be framed in the parse-plan-execute framework proposed above. In particular, the chain-of-thought paradigm (Nye et al., 2021; Wei et al., 2022c) uses LLMs to perform each of the three steps. Program-aided language models (Gao et al., 2023; Chen et al., 2022; Lyu et al., 2023) combine the parsing and

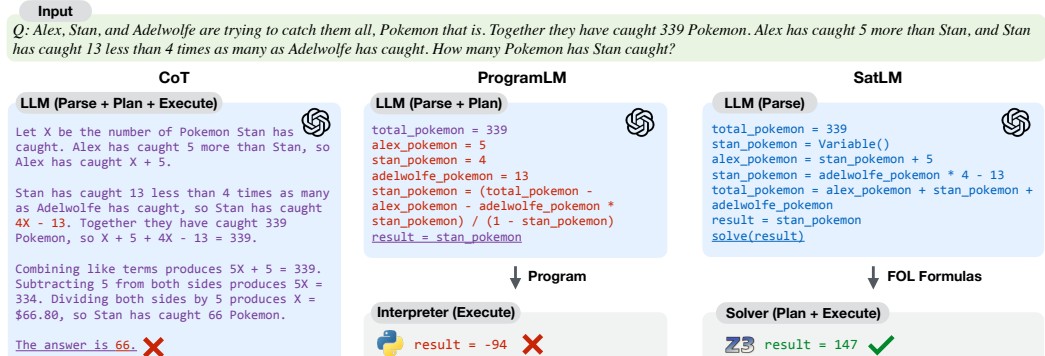

Figure 2: Exemplar specifications for arithmetic reasoning problems generated by different approaches. CoT makes errors when parsing an equation; PROGLM produces an incorrect reasoning chain (both errors are highlighted in red). By only using the LLMs to generate declarative specifications and relying on a solver to handle the reasoning, SATLM generates the correct answer.

planning steps to use an LLM to derive a program that corresponds to the plan.[3] The final execution step is then performed by using the interpreter of the underlying programming language to derive the final answer. In contrast to these approaches, our work uses an LLM only to perform the parsing step, which is an easier problem for LLMs than planning.

We show a concrete example comparing CoT and PROGLM with our approach in Figure 2. CoT performs all three steps with the LLM. For instance, "*Alex has caught $X + 5$*" in the output corresponds to "*Alex has caught 5 more than Stan*" in the NL input (parsing). Later, CoT decides how to solve for the variable $X$ with "*Combining like terms ...*" (planning). At the same time, it also derives the equation "$5X = 334$" directly in its generation (execution). However, CoT incorrectly uses the same $X$ in the equation "$X + 5$" and "$4X - 13$", when it is supposed to be different. (Note that $4X - 13$ would be correct if Stan and Adelwolfe's roles in the corresponding NL clause were reversed.) By allowing the LLM to focus only on translation, we find a lower incidence of this kind of error, in addition to eliminating planning errors. Notably, planning errors are **not** addressed by PROGLM, which does not use programmatic manipulation at this stage. Different from PROGLM, SATLM only parses the information provided in the input question, passes the parsed formulas to a solver for both planning and execution, and obtains the correct result.

## 3   SAT-Aided Language Models using Declarative Prompting

### 3.1   Declarative Prompting

We use few-shot prompting to generate the specification $s_{\text{test}}$ for the test input $x_{\text{test}}$. Specifically, we include few-shot demonstrations $(x_i, s_i)_{i=1}^k$ in the prompt, append test input $x_{\text{test}}$ after the prompt, and let the LLM complete the specification for $x_{\text{test}}$, i.e., $s_{\text{test}} \sim p(x_{\text{test}} \mid x_1, s_1, \ldots, x_k, s_k)$.

We show an example specification for a logical reasoning task in Figure 1, and an example specification for an arithmetic reasoning task in Figure 2. Observe that in both examples, our SAT formulas (i.e., the logical formulas of $[z_1, \ldots, z_n]$ in $\Sigma_{LF}$) are written as code following Python syntax, while the natural language in $\Sigma_{NL}$ is written using comment syntax. We found that including the language here as comments was useful to improve the fidelity of the translation. Our declarative prompts also use meaningful variable names and descriptive comments following the style of prompts in prior work (Gao et al., 2023; Lyu et al., 2023). Finally, we use Python rather than a specialized DSL to be more congruent with our models' pretraining data (Ouyang et al., 2022; Chen et al., 2021). See Appendix F for more details on the SAT specification.

---

[3]This is true for "faithful chain-of-thought" as well (Lyu et al., 2023). This paper describes a breakdown of the process into "translation" and "solving" stages, where the translation step corresponds to both our parsing and planning stages. The solver used in that approach for tasks like CLUTRR does not do additional planning, but merely executes the steps outlined in CoT. In addition, their approach uses Python for execution, whereas ours uses SAT and Z3 as the unifying solving framework.

## 3.2 Solving with a SAT Solver

**SAT problem** A SAT problem is a triple $\mathcal{P} = (\Phi, \mathcal{T}, Q)$ where $\Phi$ is a set of first-order logic formulas in some theory $\mathcal{T}$[4] and $Q$ is the query of interest. We use $\texttt{Variable}(\mathcal{P})$ to denote the free variables in $\Phi$. $Q$ contains only variables in $\texttt{Variable}(\mathcal{P})$. An example SAT problem is $\mathcal{P} = (\{x + y = 3, x - y = 1\}, \mathcal{T}_E \cup \mathcal{T}_{\mathbb{Z}}, x - 2)$, where $\mathcal{T}_E \cup \mathcal{T}_{\mathbb{Z}}$ indicates that only equality and linear arithmetic operations on integers are allowed in the formulas.

Many NL reasoning tasks in the literature can be formulated as SAT problems and solved using an off-the-shelf solver. For **arithmetic reasoning**, the SAT formulas $\Phi$ are equations encoding the relationships between variables, and $t$ specifies the target variable asked in the question (see Figure 1). For **logical reasoning**, $\Phi$ encodes preconditions and $t$ specifies the target statement posed by the question. We also show that symbolic reasoning, regex synthesis, and other problems involving reasoning over arrays or strings can be handled in this framework.

Unlike prior work such as Faithful CoT (Lyu et al., 2023) that uses task-specific formulations and task-specific solvers for different problem types, all the tasks in this paper are formulated as general SAT instances that can be solved by a single solver (as described later in this section).

**Parsing NL to a SAT problem** Recall that we obtain a specification $s_{\texttt{test}}$ from a test NL task $x_{\texttt{test}}$. To derive the SAT problem $\mathcal{P}_{\texttt{test}} = (\hat{\Phi}_{\texttt{test}}, \mathcal{T}_{\texttt{test}}, \hat{Q}_{\texttt{test}})$ from $s_{\texttt{test}}$, we extract the constraints $\hat{\Phi}_{\texttt{test}}$ and the target expression $\hat{Q}_{\texttt{test}}$ (marked by $\texttt{solve}$ in our prompt) by taking all the $z_i$ in $\Sigma_{LF}$ of $s_{\texttt{test}}$. We identify the theory $\mathcal{T}_{\texttt{test}}$ by analyzing the formulas in $\hat{\Phi}_{\texttt{test}}$.

**Solving the SAT problem** Given the SAT problem $\mathcal{P}$, we invoke an automated theorem prover (such as the Z3 SMT solver (De Moura and Bjørner, 2008) used in our implementation) to obtain a model $M$ that maps each free variable $v \in \texttt{Variable}(\mathcal{P})$ to a concrete value under theory $\mathcal{T}$. The final answer is obtained by substituting each free variable $v_i$ in $\hat{Q}$ with $M[v_i]$. For example, given the problem $(\{x + y = 3, x - y = 1\}, \mathcal{T}_E \cup \mathcal{T}_{\mathbb{Z}}, x - 2)$, we ask the solver to find a solution to the constraint $x + y = 3 \wedge x - y = 1$ in the theory $\mathcal{T}_E \cup \mathcal{T}_{\mathbb{Z}}$, which yields $x = 2$ and $y = 1$. Then, to obtain the final answer, we substitute $x$ by 2 in the target expression $x - 2$ to obtain the result $2 - 2 = 0$.

**Feedback signals from the solver** Given a set of $\hat{\Phi}$ specified in $\mathcal{P}$, the SAT solver will try to search for a satisfying assignment $M$ which satisfies all constraint formulas in $\hat{\Phi}$. If the solver succeeds in finding such an assignment within a certain time limit, it will use $M$ to evaluate the query $\hat{Q}$ and return the final result, otherwise it is a timeout. However, the solver may fail to find a solution for problematic $\mathcal{P}$ and provide feedback in one of the following types: (1) **error in execution** (ERROR) caused by invalid formulas (e.g., syntax errors) or time-out; (2) **unsatisfiable formulas** (UNSAT), caused by conflicting formulas in the $\hat{\Phi}$ (e.g. $\hat{\Phi} = \{x = y + 1, y = x + 1\}$) (no feasible solution); (3) **ambiguous formulas** (AMBIG), caused by the existence of multiple feasible solutions (e.g. $\hat{\Phi} = \{x = y + 1, x > 0\}$). Examples of SAT formulas leading to UNSAT or AMBIG can be found in Appendix G.

Unlike the executor used in PROGLM that can only detect errors in code execution, SAT solver can spot UNSAT and AMBIG in addition to ERROR. We show this unique characteristic allows our SATLM to abstain from potentially incorrect predictions much more effectively compared to PROGLM in the selective prediction setting (El-Yaniv and Wiener, 2010) (Section 4.4).

## 4 Experiments

### 4.1 Setup

**Tasks** Our work investigates 8 datasets covering 4 tasks, with a focus on arithmetic reasoning and logical reasoning tasks. We list all dataset statistics in Appendix A. For arithmetic reasoning, we use GSM (Cobbe et al., 2021), GSM-SYS, and ALGEBRA (He-Yueya et al., 2023). GSM-SYS

---

[4]The theory defines the meaning of some of the symbols used in the formula. For example, in the theory of linear arithmetic, axioms of the theory give meaning to operators like addition, less than, etc.

Table 1: Comparison of our approach (SATLM) against standard prompting (directly predicting the answer), COT and PROGLM. Certain settings are not applicable (marked as −) as described in Appendix B. With greedy decoding, SATLM outperforms COT and PROGLM on all datasets by a substantial margin except for GSM, where it is on par with PROGLM. With self-consistency decoding, SATLM is consistently better than PROGLM, giving SoTA accuracy on LSAT and BOARDGAMEQA.

| | GSM-SYS | GSM | ALGE | LSAT | BOARD | CLUTRR | PROOF | COLOR | REGEX |
|---|---|---|---|---|---|---|---|---|---|
| | *code-davinci-002 (greedy decoding)* | | | | | | | | |
| STANDARD | 21.0 | 22.2 | 45.9 | 22.0 | 44.6 | 41.2 | 76.6 | 75.7 | − |
| COT | 46.5 | 62.7 | 53.6 | 23.5 | 60.7 | 40.8 | 80.1 | 86.3 | − |
| PROGLM | 43.4 | **72.7** | 52.3 | − | − | 58.9 | 83.7 | 95.1 | 39.1 |
| SATLM | **69.4** | 71.8 | **77.5** | **35.0** | **79.4** | **68.3** | **99.7** | **97.7** | **41.0** |
| | *code-davinci-002 (self-consistency decoding)* | | | | | | | | |
| COT | 56.1 | 77.3 | 64.9 | 23.1 | 62.8 | 45.7 | 88.7 | 90.6 | − |
| PROGLM | 53.4 | 82.4 | 57.7 | − | − | 71.9 | 91.2 | 98.0 | 56.5 |
| SATLM | **80.9** | **84.8** | **90.9** | **37.4** | **80.7** | **80.1** | **99.7** | **99.4** | **59.7** |

is a special subset of GSM containing examples that are paired with human-annotated solutions involving systems of equations (see Appendix A for more details). For logical reasoning, we use LSAT (Zhong et al., 2022), BOARDGAMEQA (Kazemi et al., 2023), CLUTRR (Sinha et al., 2019), and PROOFWRITER (Tafjord et al., 2021). For BOARDGAMEQA, we report the average performance on the three data splits (depth 1 to depth 3).

For CLUTRR, we use exemplars requiring up to 3 intermediate steps but evaluate on test examples requiring up to 10 intermediate steps (Sinha et al., 2019), following past work (Lyu et al., 2023). For PROOFWRITER, we evaluate on the most challenging examples requiring depth-5 proofs (Tafjord et al., 2021). For symbolic reasoning, we use Colored Object (COLOR) from BIG-bench (et al., 2022) as an exemplar task. This task can be abstracted as finding elements in a list under certain constraints. We also evaluate on a regex synthesis dataset, STREGEX (Ye et al., 2020), which requires synthesizing a regex give NL description. We cast this task into synthesizing the surface form (i.e., a string) of the target regex, and use SATLM to parse NL description into constraints over the string.

**Baselines** We compare SATLM against 3 baselines, including standard prompting (directly giving the answer), chain-of-thought prompting (COT), and executor-augmented LLMs (PROGLM). We do not compare to zero-shot baselines such as zero-shot CoT, which generally underperform few-shot CoT by a large margin on the tasks we investigate (Kojima et al., 2022).

For COT and PROGLM, we leverage prompts of existing work (Gao et al., 2023; Lyu et al., 2023; Creswell et al., 2023) whenever possible. For SATLM, we manually write prompts for the **same exemplar sets** used in COT and PROGLM to ensure a fair comparison. We note that some settings, such as PROGLM for LSAT, are not applicable. Please refer to Appendix B for more discussion of the setup, including details on the prompts we use. We also include example prompts for all the datasets in Appendix I.

**Language Models & Decoding** We conduct our main experiments and analysis on `code-davinci-002` (Chen et al., 2021), a state-of-art LLM for code and code-adjacent tasks. We evaluate the performance with both greedy decoding and self-consistency decoding (Wang et al., 2022b). Following past work (Gao et al., 2023), we use 40 samples on all datasets except for LSAT, BOARDGAMEQA, and PROOFWRITER; we use 5 samples on these datasets involving long prompts and high computation cost. For COT and PROGLM, we use a temperature of 0.7; for SATLM, we use a higher temperature of 0.9, which we find to work better.

### 4.2 Main Results

Table 1 shows the performance of our approach compared to the baselines. In general, our SAT-aided approach outperforms both COT and PROGLM by a substantial margin except on GSM with greedy decoding. We perform significance tests via bootstrap resampling, and all improvements of SATLM over PROGLM are statistically significant ($p < 0.05$).

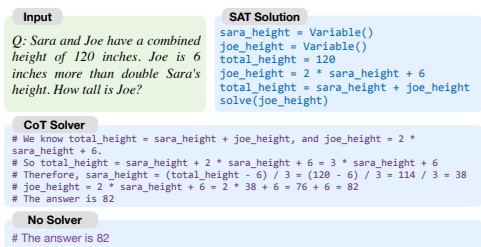

Figure 3: A variant of our approach which replaces the SAT solver with a "CoT solver" that takes the SAT problem as input and solves it in natural language.

Table 2: The performance of variants of our approach that use CoT Solver or No Solver. Using declarative prompting with CoT solver is more effective than imperative CoT prompting.

|  | GSM-SYS | GSM | CLUTRR |
|---|---|---|---|
| STANDARD | 21.0 | 22.2 | 41.2 |
| CoT | 46.5 | 62.7 | 40.8 |
| PAL | 43.4 | 72.8 | 58.9 |
| SAT$_{\text{SYMSOLVER}}$ | 69.4 | 71.7 | 68.3 |
| SAT$_{\text{COTSOLVER}}$ | 54.5 | 63.2 | 48.9 |
| SAT$_{\text{NOSOLVER}}$ | 26.6 | 23.7 | 40.7 |

The first two columns show the performance on the GSM dataset. COT and PROGLM achieve much worse performance on GSM-SYS than on GSM, indicating that GSM-SYS is a challenging subset. On this subset, SATLM achieves 69.4% and 80.9% with greedy decoding and self-consistency decoding, surpassing both PROGLM and COT more than by 20%. On the original GSM dataset, the SATLM model has a slightly lower accuracy than PROGLM with greedy decoding, but outperforms it with self-consistency decoding by 2.4%; we provide detailed analysis accounting for the differences later in this section. This self-consistency accuracy of 84.8% even exceeds recent work that uses the full training set with `code-davinci-002` (82.3% in DIVERSE (Li et al., 2022b); 84.5% in LEVER (Ni et al., 2023)). On ALGEBRA, a challenging dataset of math problems extracted from algebra textbooks, SATLM also outperforms COT and PROGLM by more than 20%.

On LSAT, CLUTRR, PROOFWRITER, and COLOR, SATLM consistently achieves the best performance with either greedy decoding or self-consistency decoding. SATLM also sets the new SoTA on both LSAT and BOARDGAMEQA, surpassing previous models that are trained on the full training set. Specifically, SATLM elevates the SoTA from 30.9% (Zhong et al., 2022) to 37.4% on LSAT and from 73.9% (Kazemi et al., 2023)) to 80.7% on BOARDGAMEQA. See Appendix E for detailed performance breakdown on depth 1-3.

In the regex synthesis domain, with greedy decoding, directly translating natural language descriptions to regexes (PROGLM) achieves 37.1%, whereas using declarative prompting achieves 44.0%. With self-consistency, we surpass the previous SoTA performance of 55.6% (Ye et al., 2021).

## 4.3 Impact of SAT Solver & Declarative Prompting

We conduct analysis to isolate the effectiveness of the two key components, the SAT solver and declarative prompting. Specifically, we test a variant of our approach that still uses declarative prompting but then solves the equations in natural language with CoT rather than using the symbolic solver (see Figure 3). Essentially, the LLM itself carries out planning and execution. This experiment helps isolate the benefits of the solver, which will compute an answer without making any mistakes, from the benefits of the declarative formulation. We also compare to prompting LLMs to directly give the answer (NOSOLVER).

**Impact of Symbolic Solver** As shown in Table 2, completely ablating the solver and directly predicting the answer (SAT$_{\text{NOSOLVER}}$) only yields performance that is on par with STANDARD. Interestingly, SAT$_{\text{COTSOLVER}}$ can solve more SAT problems than NOSOLVER. This partially reflects the effectiveness of CoT and partially reflects the fact that many dataset instances require relatively simple planning and execution, allowing pure forward reasoning to solve them. However, using a symbolic solver

Table 3: Fraction of planning errors (incorrect reasoning chains) and execution errors (numeric errors) made by COTSOLVER.

|  | GSM-SYS | GSM | CLUTRR |
|---|---|---|---|
| PLAN ERR | 72.5 | 42.5 | 47.5 |
| EXEC ERR | 27.5 | 57.5 | 52.5 |

(SAT$_{\text{SYMSOLVER}}$), which guarantees correct planning and execution, leads to further improvements.

Table 5: Analysis of accuracy and execution status of SATLM and PROGLM. We present the fraction of tasks solved correctly or incorrectly in GSM-SYS, GSM, and CLUTRR, along with the breakdown of feedback from the solver. SATLM generally makes fewer predictions than PROGLM (ANSWERED), but more frequently makes correct predictions when it returns an answer (SELECTIVE ACC) *and* gives a higher absolute number of correct predictions on GSM-SYS and CLUTRR.

| | GSM-SYS | | GSM | | CLUTRR | |
|---|---|---|---|---|---|---|
| | PROGLM | SATLM | PROGLM | SATLM | PROGLM | SATLM |
| CORRECT | 43.3 | 69.4 | 72.7 | 71.8 | 58.9 | 68.3 |
| INCORRECT | 52.5 | 20.6 | 25.7 | 21.2 | 21.0 | 7.7 |
| ERROR | 4.2 | 2.6 | 1.6 | 2.1 | 20.1 | 3.5 |
| UNSAT | – | 2.4 | – | 1.5 | – | 15.5 |
| AMBIG | – | 5.0 | – | 3.4 | – | 5.0 |
| ANSWERED | 95.8 | 90.0 | 98.4 | 93.0 | 79.9 | 76.0 |
| SELECTIVE ACC | 45.2 | **77.1** | 73.8 | **77.2** | 73.7 | **89.9** |

We manually analyzed 40 cases where the symbolic solver yields the correct answer but $\text{SAT}_{\text{CoTSolver}}$ fails to solve them. We categorized the errors as planning errors, where the reasoning chains are incorrect, and execution errors, where the reasoning chains are correct but computations are incorrect (see Appendix H for examples). Table 3 shows that most errors by $\text{SAT}_{\text{CoTSolver}}$ are planning errors, especially on GSM-SYS which requires solving complex system of equations.

**Impact of Declarative Prompting** Table 2 also shows that decoupling parsing and planning/solving is still useful, even when not using a symbolic solver: $\text{SAT}_{\text{CoTSolver}}$ outperforms CoT by 7.9%, and 8.1% on GSM-SYS and CLUTRR, respectively. We note that $\text{SAT}_{\text{CoTSolver}}$ can be viewed as a two-stage CoT prompting strategy, with a prompt showing that the first step is to formulate declaratively, then the next step is to solve.

We hypothesize that parsing a question into declarative formulas is more straightforward than parsing it into an imperative solving procedure. To evaluate this hypothesis, we use log likelihood of the generated tokens to assess how straightforward the translation is, as higher

Table 4: Log likelihood (unnormalized / normalized) of the generated sequences (with greedy decoding) of PROGLM and SATLM on three datasets. Better log likelihood indicates higher LLM confidence in the parsing stage.

| | GSM-SYS | GSM | CLUTRR |
|---|---|---|---|
| PAL | -9.5/-6.9$_{10^{-2}}$ | **-9.2/-6.0**$_{10^{-2}}$ | -3.1/-8.5$_{10^{-3}}$ |
| SAT | **-8.5/-5.9**$_{10^{-2}}$ | -9.7/-6.2$_{10^{-2}}$ | **-2.0/-7.9**$_{10^{-3}}$ |

log-likelihood typically indicates the outputs are more fluent to LLMs, a connection demonstrated in recent literature (Gonen et al., 2022; Ye and Durrett, 2023). We show both unnormalized (total) and normalized log likelihood in Table 4. On GSM-SYS and CLUTRR where SATLM outperforms PROGLM, its generated outputs are also associated with higher likelihood.

## 4.4 Advantages of SAT in Selective Prediction

A SAT solver may not always return an answer, particularly if there are parsing errors from the question. We show that this is an advantage of SATLM: these errors allow us to abstain from making likely incorrect predictions. Example outputs leading to different errors can be found in Appendix G.

Table 5 shows the fraction of correct predictions and incorrect predictions when the program or SAT solver successfully returns an answer as well as the fraction of different types of feedback signals. We report the fraction of questions *answered* as well as *selective accuracy*, defined by the fraction of overall accuracy (% of correct answers) normalized by coverage (% of answered problems). SATLM makes fewer predictions on all three datasets compared to PROGLM, as it can trigger both UNSAT and AMBIG errors. However, SATLM's selective accuracy is consistently better than PROGLM's, especially on GSM-SYS (77% vs 45%). As a result, SATLM's overall performance is significantly better than PROGLM on GSM-SYS and CLUTRR, even when making fewer predictions.

We note that on GSM, SATLM has slightly lower coverage but higher selective accuracy compared to PROGLM. This explains why SATLM lags behind PROGLM with greedy decoding but outperforms PROGLM with self-consistency decoding (Table 1). By drawing multiple samples, SATLM can increase its coverage and achieve higher accuracy than PROGLM since its predictions are more accurate.

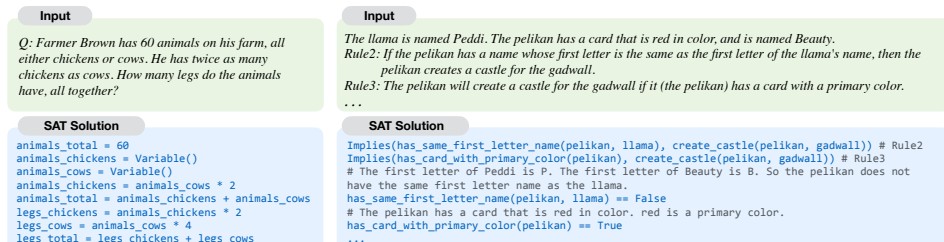

Figure 4: Examples outputs from GSM (left) and BOARDGAMEQA (right) show that LLMs can perform commonsense reasoning while parsing.

## 4.5 Analysis

**LLMs Can Perform Commonsense Reasoning While Parsing**   There are many problems that do not state premises or constraints in a completely explicit way. Figure 4) shows two examples where commonsense inferences are required during parsing. For example, on the left, the model must recognize that *animals* refers to the chickens and cows collectively. Similarly, knowing that red is a primary color is needed to successfully apply rules on BOARDGAMEQA (right). We observe from the outputs in both cases that LLMs are capable of implicitly performing commonsense reasoning and produce correct logical formulas in the parsing step. As shown in Table 1, SATLM exhibits strong performance on BOARDGAMEQA, a dataset which requires this implicit background knowledge.

**Results Across Different Language Models**
In addition to the main LLM used in our work, `code-davinci-002`, we further test whether SATLM can generalize to other LLMs. We choose `gpt-3.5-turbo` (0613 version), `text-davinci-003`, and `code-davinci-001`. `gpt-3.5-turbo` is optimized for chat. `text-davinci-003` is an LLM pretrained on NL, and tuned to align with human feedback (Ouyang et al., 2022). `code-davinci-001` is also an LLM pretrained on code, but less capable compared to `002`. As shown in Table 6, SATLM is better than PROGLM on the arithmetic reasoning and logical reasoning datasets except for GSM across these three LLMs. The trend is congruent with the results on `code-davinci-002` (Table 1), which suggests the approach's general applicability across different LLMs, regardless of their varying capabilities.

Table 6:   Results on `gpt-3.5-turbo`, `text-davinci-003`, and `code-davinci-001`. The effectiveness of SATLM can generalize across LLMs.

|  | GSM-SYS | GSM | LSAT | CLUTRR | PROOF |
|---|---|---|---|---|---|
| *gpt-3.5-turbo (greedy decoding)* | | | | | |
| CoT | 44.8 | 74.4 | 23.9 | 41.2 | 82.3 |
| PROGLM | 51.2 | **77.9** | – | 45.9 | 76.4 |
| SATLM | **63.4** | 76.4 | **30.0** | **50.6** | **96.4** |
| *text-davinci-003 (greedy decoding)* | | | | | |
| CoT | 42.8 | 62.5 | 21.7 | 34.5 | 83.5 |
| PROGLM | 40.4 | **71.7** | – | 41.2 | 83.7 |
| SATLM | **63.6** | 70.3 | **30.4** | **58.2** | **99.7** |
| *code-davinci-001 (greedy decoding)* | | | | | |
| PROGLM | 15.5 | **35.6** | – | 22.2 | 63.8 |
| SATLM | **16.5** | 34.2 | 19.6 | **30.2** | **86.6** |

**Sensitivity to Different Exemplar Sets**   We test whether the advantages of SATLM is sensitive to different sets of exemplars. We experiment with 3 sets of exemplars on `code-davinci-002`. As shown in Table 7, SATLM consistently outperforms PROGLM by a large margin on GSM-SYS and CLUTRR, and achieves comparable performance on GSM. The results suggest the effectiveness of our approach is insensitive to varying the choice of exemplars.

Table 7: The performance of PROGLM and SATLM with varying exemplar sets. SATLM consistently outperforms PROGLM on GSM-SYS and CLUTRR.

|  |  | GSM-SYS | GSM | CLUTRR |
|---|---|---|---|---|
| Set1 | PROG | 43.4 | **72.7** | 58.9 |
|  | SAT | **69.4** | 71.8 | **68.3** |
| Set2 | PROG | 41.4 | **72.5** | 59.0 |
|  | SAT | **71.8** | 71.3 | **67.9** |
| Set3 | PROG | 37.1 | **70.3** | 57.2 |
|  | SAT | **66.7** | 70.0 | **68.0** |

## 5   Related Work

Our work is built on top of few-shot prompting (Brown et al., 2020), which has proven effective on a wide range of tasks (Wei et al., 2022b; Liu et al., 2023b; Gehrmann et al., 2021; Reif et al., 2022; Wei et al., 2022a; Sanh et al., 2022). In particular, we focus on improving LLMs on reasoning tasks, which are challenging for language models even with recent

developments (Marcus, 2020; Garcez and Lamb, 2023). Various techniques have been proposed for improving reasoning abilities (Nye et al., 2021; Zhou et al., 2022; Kojima et al., 2022; Khot et al., 2022; Fu et al., 2022; Wang et al., 2022a; Li et al., 2022a; Lyu et al., 2023). They largely follow a chain-of-thought (Wei et al., 2022c) or scratchpad (Nye et al., 2021) paradigm. Among them, our work is the most related to the line of work that generates imperative programs to be executed by a symbolic executor, such as a Python interpreter (Gao et al., 2023; Chen et al., 2022) or domain-specific executors (Lyu et al., 2023). In this work, we propose a different paradigm that parses NL problems into declarative SAT problems and offloads the solving procedure to a SAT solver.

Previous work has also explored equipping LLMs with other tools, including search engines (Yu et al., 2023; Schick et al., 2023), calculators (Cobbe et al., 2021; Chowdhery et al., 2022), or other domain-specific special modules (Schick et al., 2023; Demeter and Downey, 2020). A line of work focuses on using program-related tools such as program executors (Poesia et al., 2022), program analysis tools (Jain et al., 2022), and synthesis tools (Rahmani et al., 2021) to enhance the quality of the generated code. Our works further explores improving LLMs with SAT solvers.

Concurrent work explores the intersection of LLMs and planning, parsing planning problems into PDDL descriptions and leveraging a classical planner to produce the plan (Liu et al., 2023a). Our work differs in that we use the SAT formulation to solve general reasoning tasks, including arithmetic reasoning and logical reasoning, which cannot be specified in PDDL.

Also concurrently, He-Yueya et al. (2023) combine LLMs and symbolic solvers for solving math problems. However, this work *only* focus on arithmetic reasoning tasks and employs a math-specific symbolic solver (PySym). Our work takes a more general approach by formulating the problem within the scope of first-order logic and therefore is domain-agnostic. We also provide results of SATLM on the ALGEBRA dataset collected by He-Yueya et al. (2023) in Appendix D.

## 6    Conclusion & Limitations

We have presented a framework for satisfiability-aided language models, casting a wide range of reasoning tasks into SAT problems under a unified formulation. We use an LLM to parse an NL query into a declarative specification and leverages a SAT solver to derive the final answer. Evaluation results on 8 datasets spanning 4 tasks across several LLMs demonstrate the effectiveness of our approach over program-aided language models.

**Limitations**   Our framework parses an NL problems into a set of declarative formulas. The NL description of some problems may already be more compatible with an imperative solving procedure, and our approach is likely to be less effective in these cases (e.g., SATLM slightly lags PROGLM on GSM). Future research can explore an integration or ensemble of these two prompting styles for more flexible reasoning.

SATLM heavily relies on the SAT solver and inherits some limitations of the SAT solver itself, such as computational cost when dealing with complex formulas involving quantifiers or nonlinear arithmetic. Moreover, SAT solvers can be limited by the expressiveness of the underlying theory, as not all theories can be easily encoded in first-order logic. Nevertheless, the wide range of tasks that we can instantiate our SATLM framework on shows its general applicability.

Our current approach parses a problem into a SAT specification, runs the solver, and returns the answer in a one-round fashion. One can imagine that unsatisfiable formulas or ambiguous formulas could be improved by re-prompting the model to improve the specification based on the exception signals, as explored in concurrent work for other problems (Paul et al., 2023; Madaan et al., 2023; Chen et al., 2023). We believe this is an exciting direction for future work.

## Acknowledgments

Thanks to anonymous reviewers for their helpful feedback. This work was partially supported by National Science Foundation under Grants No.2145280, No.1918889, No.1762299, No.2210831 and the NSF AI Institute for Foundations of Machine Learning (IFML). We would also like to thank

authors of PAL (Gao et al., 2023) and Faithful CoT (Lyu et al., 2023) for providing the prompts used in the baselines.

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

# A  Detailed Statistics of Datasets

We show the statistics of all the datasets used in our paper in Table 8.

For CLUTRR, we follow the setting in FAITHFULCOT (Lyu et al., 2023): we construct the prompt using exemplars requiring 2-3 reasoning steps and test whether the model can generalize to examples requiring up to 10 steps. We used the pre-processed test data consisting of 1,042 test examples from past work (Lyu et al., 2023).

For PROOFWRITER, we use the closed world assumption setting, following past work (Creswell et al., 2023). We construct our test set by randomly sampling a subset of 1,000 examples (out of 10,000) from the test split of depth-5 setting, the most challenging setting.

For STREGEX, we merge the test and test-E split (see Ye et al. (2020)) to form a test set consisting of 996 examples in total.

Table 8: Number of few-shot exemplars, number of test examples and license for the datasets used in our paper.

|  | # Shot | # Test | License |
|---|---|---|---|
| GSM (Cobbe et al., 2021) | 8 | 1,319 | MIT license |
| GSM-SYS | 8 | 547 | MIT license |
| ALGEBRA (He-Yueya et al., 2023) | 8 | 222 | Creative Commons Attribution Share Alike 4.0 |
| LSAT (Zhong et al., 2022) | 8 | 230 | MIT license |
| BOARDGAMEQA (Kazemi et al., 2023) | 5 | 3,000 | CC BY 4.0. |
| CLUTRR (Sinha et al., 2019) | 8 | 1,042 | Attribution-NonCommercial 4.0 |
| PROOFWRITER (Tafjord et al., 2021) | 4 | 1,000 | CC BY 4.0. |
| COLOREDOBJECT (BIG-BENCH) | 3 | 2,000 | Apache 2.0 |
| STRUCTUREDREGEX (Ye et al., 2020) | 8 | 996 | MIT license |

**GSM-SYS Dataset**  We construct GSM-SYS, a special subset consisting of 547 examples extracted from GSM. Specifically, we filter the entire GSM dataset (train split and test split) to find examples whose human-annotated explanations involve a system of equations, using patterns like *"let [letter] be"*, *"assume [letter] be"* and *"[number][letter]"*. We manually inspected 10% of the examples and found 80% of those samples did involve systems of equations in the explanation. We refer to this more challenging dataset as GSM-SYS.

# B  Details of the Prompts

In general, we leverage COT prompts and PROGLM prompts from existing work whenever available, and manually write SATLM prompts for the **same exemplar sets**. Prompt examples for all datasets can be found in Appendix I.

For **GSM and GSM-SYS**, we adapt the original COT prompt and PROGLM prompt used in program-aided language models (Gao et al., 2023). Specifically, we replace one random exemplar in the original prompt with another exemplar sampled from GSM-SYS. This is to improve the performance of COT and PROGLM on GSM-SYS, as the original exemplar set achieves suboptimal performance for GSM-SYS. Our adapted COT and PROGLM prompts achieve better performance compared to the original ones on both GSM and GSM-SYS (see Appendix C for details).

For **LSAT**, we randomly sample 8 exemplars and write prompts for COT and SATLM. We note that LSAT is a particularly challenging task: we tried 3 CoT prompts written by 3 different authors of our paper, which all led to around 20% accuracy. Similar results are reported in other work (Liang et al., 2022; Ribeiro et al., 2023). In addition, we only report COT results, leaving out PROGLM. This decision is due to the fact that PROGLM uses Python as its program interpreter. While Python is a general-purpose programming language, it does not provide native support for formal logic reasoning, including essential components like logical inference rules and manipulation of logical formulas. Solving problems from LSAT requires strategies like proof by contradiction (see Appendix I for a detailed example), which we see no way to represent in the PROGLM framework and is not addressed in prior work.

BOARDGAMEQA contains problems requiring 1-3 steps of reasoning. We sample 5 exemplars from the training set of depth 1 and depth 2 to construct the prompts for evaluation on the test sets of depth 1 and depth 2, respectively. We used the 5 exemplars of depth 2 to construct the prompt for test set of depth 3, as using exemplars of depth 3 would lead to prompts that exceed the context window size of our LLMs. Similarly, we only report CoT results as the baselines, leaving out PROGLM for BOARDGAMEQA. We use the proofs provided by the authors to construct the CoT prompts and manually annotate the SAT specifications to construct the SATLM prompts.

For CLUTRR, we use the CoT prompt and PROGLM prompt provided in FAITHFULCOT (Lyu et al., 2023). For PROOFWRITER, we use the CoT prompt from SELECTION-INFERENCE (Creswell et al., 2023), and adapt it to form the PROGLM prompt. We use the CoT prompt and PROGLM from PAL (Gao et al., 2023) for COLORED OBJECT.

The task of STRUCTUREDREGEX, a regex synthesis dataset, is to parse natural language descriptions to regexes. This is not a typical reasoning dataset, and there is no CoT prompt for this dataset. We randomly sample 8 exemplars and annotate the prompt for PROGLM and SATLM. In this setting, PROGLM directly translates NL descriptions into regexes (which are essentially programs), whereas SATLM parses an NL description into a set of constraints over the surface form of the regex. Note that this dataset provides *multimodal* specifications of regexes, featuring both NL descriptions and examples. The I/O examples can be used to reject synthesized regexes if they do not accept or reject the correct examples. When we report results for self-consistency inference, we follow past work (Ye et al., 2021) and filter out incorrect outputs using the I/O examples provided in the dataset (Ye et al., 2020). This setting therefore checks consistency with something other than the model itself, but uses a similar computation budget as self-consistency, so we group it with those results.

## C   Performance of Original CoT and PROGLM Prompts on Arithmetic Reasoning Datasets

Table 9: Performance of different approaches using our adapted exemplar set and the original exemplar set used in CoT and PAL.

|  | ADAPTED (OURS) | | ORIGINAL | |
|  | GSM-SYS | GSM | GSM-SYS | GSM |
| --- | --- | --- | --- | --- |
| CoT | 46.5 | 62.7 | 35.7 | 62.4 |
| PROGLM | 43.4 | **72.7** | 36.1 | **71.7** |
| SATLM | **69.4** | 71.8 | **66.7** | 70.9 |

Recall that we construct our arithmetic reasoning prompt used in Table 1 by replacing one random exemplar in the original prompt used in CoT and PROGLM with an random example from GSM-SYS. We show the performance of CoT, PROGLM, and our SATLM in Table 9 using our adapted exemplar set and original exemplar set in Table 9.

Our adaptation significantly improves the performance of CoT and PROGLM on GSM-SYS, and slightly improves the performance on GSM. Furthermore, we still see that SATLM outperforms both CoT and PROGLM by a large margin on GSM, using either our adapted set or the original set.

## D   Extended Discussion on Concurrent Work

Table 10: Performance of different approaches on ALGEBRA.

|  | ALGEBRA | GSM |
| --- | --- | --- |
| CoT | 53.6 | 62.4 |
| PROGLM | 52.3 | 72.7 |
| SATLM (Ours) | 77.5 | 71.8 |
| MATHSYM (He-Yueya et al., 2023) | 76.3 | 69.4 |

Similar to our work, He-Yueya et al. (2023) proposes to solve arithmetic reasoning problems by parsing the problem into a set of variables and equations and using an external solver to derive the

final answer. While their formalization is restricted to arithmetic problems, we use SAT problems encoded with first-order logical formulas, which unify a wide range of reasoning tasks.

In addition, we also evaluate our approach on the ALGEBRA dataset in He-Yueya et al. (2023), which consists of 222 examples from Algebra textbooks. We note that the results between ours and MATHSYM are not directly comparable, as MATHSYM picks a different exemplar set. As shown in Table 10, ALGEBRA is more challenging than GSM, and SATLM outperforms PROGLM and COT by more than 20%.

## E  Detailed Performance on the BOARDGAMEQA Dataset

Table 11: Detailed performance on the BOARDGAMEQA dataset.

| | DEPTH 1 | DEPTH 2 | DEPTH 3 | AGGREGATED |
|---|---|---|---|---|
| | *code-davinci-002 (greedy decoding)* | | | |
| STANDARD | 52.5 | 42.8 | 38.5 | 44.6 |
| COT | 64.7 | 60.8 | 56.5 | 60.1 |
| SATLM | **87.6** | **81.7** | **69.0** | **79.4** |
| | *code-davinci-002 (self consistency decoding)* | | | |
| COT | 65.9 | 63.4 | 59.0 | 62.8 |
| SATLM | **88.0** | **84.2** | **70.1** | **80.8** |

Table 11 shows the performance breakdown on depths 1-3 of the BOARDGAMEQA dataset. SATLM outperforms COT by a substantial margin across all depths. The performance of all approaches decreases as the depth increases.

## F  Details of the SAT Specification

To better utilize the parametric knowledge that LLMs have acquired from pretraining on vast amount of code data, our work uses a specification that largely follows and simplifies the syntax for specifying constraints used in z3py.[5]

---

**Example SAT Specification**

```
x = Variable() # declare a variable
People = [Alice, Bob] # declare enum set
Cities = [Austin, Boston] # declare enum set
Food = [Apple, Banana] # declare enum set

visit = Function(People, Cities) # declare function
eats = Function(People, Food) # declare function

visit(Alice) != visit(Bob) # logic
ForAll(x: People, Implies(visit(x) == Austin, eats(x) == Banana)) # quantifier
```

Figure 5: Example of our SAT specification. The syntax is largely the same as that for specifying constraints in z3py.

We give an example specification in Figure 5 demonstrating the synax for different types of statements. See Figure 1, Figure 2, and Appendix I for more examples. These formulas are close to the actual python code formulas used by z3py but are slightly modified to be more amenable to prompting. As a result, we use a postprocessing step to form the actual Z3 input. We implemented a simple parser that transforms these formulas into actual specifications used by z3py via string transformation (using regexes). For example, we transform [ForAll(x: People, Implies(visit(x, Austin), eats(x, Banana)))] into [x = Variable(People) ForAll([x], Implies(visit(x) == Austin, eats(x) ==

---

[5]https://z3prover.github.io/api/html/namespacez3py.html

`Banana))`], which is the actual z3py code. The transformed code can be executed using z3py to obtain the final answer.

# G   Examples of Error Cases of SatLM

## G.1   Examples of Unsatisfiable and Ambiguous Formulas

We show examples of formulas that are unsatisfiable or have ambiguous answers in Figure 6, Figure 7, and Figure 8.

---

**UNSAT Formulas: GSM**

---

Q: If a rectangle has a width of 42 inches and an area of 1638, how many rectangles of the same size would reach a length of 390 inches?

```
rectangle_width = 42
rectangle_area = 1638
rectangle_length = 390
rectangle_length = rectangle_area / rectangle_width
rectangles_needed = rectangle_length / rectangle_width
result = rectangles_needed
solve(result)
```

Q: Jason's dog has a tail that's half the length of its body, and a head that's 1/6 the length of its body. If the dog is 30 inches long overall, how long is its tail?

```
dog_length = 30
tail_length = Variable()
head_length = Variable()
dog_length = tail_length + head_length
tail_length = dog_length * 1 / 2
head_length = dog_length * 1 / 6
result = tail_length
solver(result)
```

---

Figure 6: Examples of UNSAT formulas from GSM. Both examples are generated by `code-davinci-002`. UNSAT occurs when LLMs misunderstand the problem and output conflicting constraints (highlighted in red).

---

**AMBIG Formulas: GSM**

---

Q: The sum of the three numbers is 500. If the first number is 200, and the value of the second number is twice the value of the third number, find the value of the third number.

```
num1 = 200
num2 = Variable()
num3 = Variable()
num2 = num3 * 2
total = num1 + num2 + num3
result = num3
solve(result)
```

Q: The total for the sum and product of Elvie's age and Arielle's age are 131. If Elvie's age is 10, how old is Arielle?

```
elvie_age = 10
arielle_age = Variable()
sum_age = elvie_age + arielle_age
product_age = elvie_age * arielle_age
total_age = sum_age + product_age
result = arielle_age
solve(result)
```

---

Figure 7: Examples of AMBIG solutions from GSM. Both examples are generated by `code-davinci-002`. The ambiguous formulas are under-constrained due to failure in encoding certain constraints specified in the problem (highlighted in red), resulting in multiple possible answers.

| **UNSAT** Formulas: CLUTRR |
| --- |

Arlene and her husband Jeff went on a cruise. They had a wonderful time. Stephanie s father Jason loves his little princess even though she gets into a lot of trouble at school. Gloria's mother Ruth and brother Jeff were working in the kitchen, preparing soup. Stephanie, Ruth's daughter, was working outside in the garden.
Q: How is [Jason] related to [Arlene]?

```
# [Arlene] and her husband [Jeff] went on a cruise. They had a wonderful time.
relation(Arlene, Jeff) = (wife, husband)
# [Stephanie]'s father [Jason] loves his little princess even though she gets into a lot of trouble at school.
relation(Stephanie, Jason) = (daughter, father)
# [Gloria]'s mother [Ruth] and brother [Jeff] were working in the kitchen, preparing soup.
relation(Gloria, Ruth) = (daughter, mother)
relation(Gloria, Jeff) = (daughter, brother)
# [Stephanie], [Ruth]'s daughter, was working outside in the garden.
relation(Stephanie, Ruth) = (daughter, mother)
# How is [Jason] related to [Arlene]?
solve(relation(Jason, Arlene))
```

| **AMBIG** Formulas: CLUTRR |
| --- |

Kirk loves talking to his grandfather Stanley on the phone. Paul's brother, Kirk, recently made it into town. Paul and his son Michael went to look at cars. Paul ended up buying the Mustang.
Q: How is [Stanley] related to [Michael]?

```
# [Kirk] loves talking to his grandfather [Stanley] on the phone.
relation(Kirk, Stanley) = (grandson, grandfather)
# [Paul]s brother, [Kirk], recently made it into town.
relation(Paul, Kirk) = (brother, brother)
# [Paul] and his son [Michael] went to look at cars.
relation(Paul, Michael) = (father, son) note: Michael is Kirk's son, not Paul's son
# How is [Stanley] related to [Michael]?
solve(relation(Stanley, Michael))
```

Figure 8: An example of UNSAT formulas (top) and an example of AMBIG formulas (bottom) from CLUTRR. Both happen when the LLM misparses a constraint (highlighted in red). In particular, ambiguous formulas usually occur when the language description itself is highly ambiguous (see the note in the ambiguous formulas for an example).

## G.2 Examples of Incorrect Answers

We show examples where the SAT solver successfully returns an answer, but the answer is incorrect in Figure 9.

---

**Examples of Incorrect Answers: GSM**

---

Q: Martin rings the small bell 4 times more than 1/3 as often as the big bell. If he rings both of them a combined total of 52 times, how many times does he ring the big bell?

```
small_bell_rings = Variable()
big_bell_rings = Variable()
total_bell_rings = 52
small_bell_rings = big_bell_rings * 1 / 3 * 4
total_bell_rings = small_bell_rings + big_bell_rings
result = big_bell_rings
solve(result)
```

---

Q: Jonathan was sad to learn he needed 2 more toys than he already had to have enough toys to make a sorted play set five times larger than James's play set, which had 80 toys. How many toys does Jonathan currently have?

```
james_toys = 80
jonathan_toys = Variable()
jonathan_toys_needed = james_toys * 5 + 2
result = jonathan_toys_needed - 2
solve(result)
```

---

**Examples of Incorrect Answers: CLUTRR**

---

Donald and Russell are brothers. Jason is father of their father.
Q: How is [Russell] related to [Jason]?

```
# [Donald] and [Russell] are brothers.
relation(Donald, Russell) = (brother, brother)
# [Jason] is father of their father
relation(Jason, Donald) = (father, son)
relation(Jason, Russell) = (father, son)
# How is [Russell] related to [Jason]?
solve(relation(Russell, Jason))
```

---

Figure 9: Two examples of incorrect answers from GSM (top) and an example of incorrect answer from CLUTRR (bottom). This occurs when LLMs output incorrect constraints (highlighted in red) but the incorrect formulas can still derive a non-ambiguous answer.

# H Examples of Execution Errors and Planning Errors Made by COTSOLVER

We show examples of execution errors and planning errors produced by COTSOLVER in Figure 10, Figure 11, and Figure 12.

As shown in Figure 10, LLMs can make errors when executing arithmetic operators, resulting in execution errors. While introducing an interpreter (PROGLM) can help fix such execution errors, it is not effective for handling planning errors in Figure 11, which involves manipulating symbolic variables to determine a chain of arithmetic equations leading to the final answer.

We note that for CLUTRR, the atomic function is applying a set of transitivity rules that infer the relation between A and C based on the relation between A and B and the relation between B and C (e.g., A's son is B, B's daughter is C. So A's granddaughter is C). So an execution error in CLUTRR refers to incorrect outputs when applying transitivity rules (see the top section of Figure 12). The set of transitivity rules we use are provided in Lyu et al. (2023); Zhang et al. (2022a).

---

**Execution Errors of COTSOLVER: GSM**

Q: Carly is thinking about buying a wallet that costs 22 and a purse that costs 3 less than four times the cost of the wallet. What is the combined cost of both items?

```
wallet_cost = 22
purse_cost = Variable()
purse_cost = wallet_cost * 4 - 3
total_cost = wallet_cost + purse_cost
result = total_cost
# We know purse_cost = wallet_cost * 4 - 3
# So total_cost = wallet_cost + purse_cost = 22 + (22 * 4 - 3) = 22 + 87 = 109
# result = total_cost = 109
# The answer is 109
```

Q: A person borrowed an amount of money for a year at an interest rate of 12%. If the total interest is $1500, what was the original borrowed amount?

```
interest_rate = 12
interest_total = 1500
borrowed_amount = Variable()
interest_total = borrowed_amount * interest_rate / 100 result = borrowed_amount
# We know interest_total = borrowed_amount * interest_rate / 100
# So borrowed_amount = interest_total / interest_rate * 100 = 1500 / 12 * 100 = 12500
/ 12 = 1041.67
# result = borrowed_amount = 1041.67
# The answer is 1041.67
```

---

Figure 10: Examples of execution errors made by COTSOLVER on arithmetic reasoning tasks. Both examples are generated by `code-davinci-002`, which makes mistakes when calculating the final value of equations.

Q: Griffin had 24 french fries, but Kyle took 5 of them. Billy took twice as many as Kyle. Ginger gave Griffin a handful of her fries, and then Colby took from Griffin 3 less than the number of fries that Kyle had taken. If in the end Griffin had 27 fries, how many fries did Ginger give Griffin?

```
fries_initial = 24
kyle_took = 5
billy_took = kyle_took * 2
ginger_gave = Variable()
colby_took = kyle_took - 3
fries_after = 27
fries_after = fries_initial - kyle_took - billy_took + ginger_gave - colby_took
result = ginger_gave
# Solve the value of result
# We know fries_after = fries_initial - kyle_took - billy_took + ginger_gave - colby_took
# So ginger_gave = fries_after - fries_initial + kyle_took + billy_took - colby_took =
27 - 24 + 5 + 10 - 2 = 16
# result = ginger_gave = 16
# The answer is 16
```

Q: The combined age of Peter, Paul and Jean is 100 years old. Find the age of Peter knowing that Paul is 10 years older than John and that Peter's age is equal to the sum of Paul and John's age.

```
john_age = Variable()
total_age = 100
paul_age = john_age + 10
peter_age = paul_age + john_age
total_age = peter_age + paul_age + john_age
result = peter_age
# We know paul_age = john_age + 10, and peter_age = paul_age + john_age. So peter_age
= john_age + 10 + john_age = 2 * john_age + 10
# We also know total_age = peter_age + paul_age + john_age = 2 * john_age + 10 + john_age
+ 10 = 3 * john_age + 20
# Therefore, john_age = (total_age - 20) / 3 = (100 - 20) / 3 = 80 / 3 = 26.67
# peter_age = 2 * john_age + 10 = 2 * 26.67 + 10 = 53.33 + 10 = 63.33
# result = peter_age = 63.33
# The answer is 63.33
```

Figure 11: Examples of planning errors made by CoTSolver from GSM. Both examples are generated by `code-davinci-002`. In the first example, LLM fails to flip the sign of `colby_took`. In the second example, the model drops an occurrence of `john_age` when expanding out the formula. In both case, LLM fails to arrange the symbols to derive a chain of numeric equations that gives the correct answer.

---

**Execution Errors** of COTSOLVER: **CLUTRR**

---

# [James] was looking forward to dinner with his granddaughter, [Joanne].
relation(James, Joanne) = (grandfather, granddaughter)
# [Helen] went to her son [Leo]'s House
relation(Helen, Leo) = (mother, son)
# [Gene] bought gifts for his lovely sister, [Joanne]
relation(Gene, Joanne) = (brother, sister)
# [Kecia] wanted to visit an art museum, so she asked her father, [James] to take her.
relation(Kecia, James) = (daughter, father)
# [Sharon] and her sister, [Kecia], frequently bicker. However, they always make up quickly.
relation(Sharon, Kecia) = (sister, sister)
# [Leo] was so happy for his brother, [Gene], getting a promotion.
relation(Leo, Gene) = (brother, brother)
# How is [Sharon] related to [Helen]?
answer = relation(Sharon, Helen)
# Solve the problem
# Helen's son is Leo, and Leo's brother is Gene, so Helen's son is Gene.
# Helen's son is Gene, and Gene's sister is Joanne, so Helen's daughter is Joanne.
# Helen's daughter is Joanne, and Joanne's father is James, so Helen's son is James.
# Helen's son is James, and James's daughter is Kecia, so Helen's granddaughter is Kecia.
# Helen's granddaughter is Kecia, and Kecia's sister is Sharon, so Helen's granddaughter is Sharon.
# The answer is granddaughter.

---

**Planning Errors** of COTSOLVER: **CLUTRR**

---

# [Rosa] took her daughter [Melissa] to the park.
relation(Rosa, Melissa) = (mother, daughter)
# [Melissa] was sad her sister [Annie] could not join.
relation(Melissa, Annie) = (sister, sister)
# [Lisa] is having a great day at the playground with her mother [Rosa] and [Rosa]'s sister [Ruth].
relation(Lisa, Rosa) = (daughter, mother)
relation(Rosa, Ruth) = (sister, sister)
# [Rosa] asked her daughter, [Lisa], if she would like to go to a movie with her on Saturday night.
relation(Rosa, Lisa) = (mother, daughter)
# How is [Ruth] related to [Annie]?
answer = relation(Ruth, Annie)
# Solve the problem
# Rosa's daughter is Lisa, and Lisa's mother is Rosa, so Rosa's mother is Rosa.
# Rosa's mother is Rosa, and Rosa's sister is Ruth, so Rosa's aunt is Ruth.
# Rosa's aunt is Ruth, and Rosa's daughter is Melissa, so Rosa's niece is Melissa.
# Rosa's niece is Melissa, and Melissa's sister is Annie, so Rosa's niece is Annie.
# The answer is niece.

---

Figure 12: Examples of planning errors made by COTSOLVER on CLUTRR. We omit questions for brevity. Both examples are generated by `code-davinci-002`. In the first example, the model outputs an incorrect value when applying the transitivity rule marked in red (correct output should be `husband`). In the second example, the model comes up with an incorrect procedure.

# I Prompt Examples

We show one or two exemplars in the prompt for each dataset. We list prompts for PROGLM for comparison.

**Prompts for GSM and GSM-SYS**

| SATLM |
|---|

Q: Jason had 20 lollipops. He gave Denny some lollipops. Now Jason has 12 lollipops. How many lollipops did Jason give to Denny?

```
jason_lollipops_initial = 20
lollipops_given = Variable()
jason_lollipops_after = 12
jason_lollipops_after = jason_lollipops_initial - lollipops_given
result = lollipops_given
solve(result)
```

Q: Jeff bought 6 pairs of shoes and 4 jerseys for $560. Jerseys cost 1/4 price of one pair of shoes. Find the shoe's price total price.

```
shoes_num = 6
jerseys_num = 4
total_cost = 560
shoes_cost_each = Variable()
jerseys_cost_each = Variable()
shoes_cost_each * shoes_num + jerseys_cost_each * jerseys_num = total_cost
jerseys_cost_each = shoes_cost_each * 1 / 4
shoes_cost_total = shoes_cost_each * shoes_num
result = shoes_cost_total
solve(result)
```

| PROGLM from Gao et al. (2023) |
|---|

Q: Jason had 20 lollipops. He gave Denny some lollipops. Now Jason has 12 lollipops. How many lollipops did Jason give to Denny?

```
jason_lollipops_initial = 20
jason_lollipops_after = 12
denny_lollipops = jason_lollipops_initial - jason_lollipops_after
result = denny_lollipops
return result
```

Q: Jeff bought 6 pairs of shoes and 4 jerseys for $560. Jerseys cost 1/4 price of one pair of shoes. Find the shoe's price total price.

```
shoes_num = 6
jerseys_num = 4
total_cost = 560
jersey_shoes_cost_ratio = 1 / 4
shoes_cost_each = total_cost / (shoes_num + jerseys_num * jersey_shoes_cost_ratio)
shoes_cost_total = shoes_cost_each * shoes_num
result = shoes_cost_total
return result
```

Figure 13: Prompt (excerpt) used for GSM and GSM-SYS.

**Prompts for LSAT**

---

### SᴀᴛLM

---

Nine different treatments are available for a certain illness: three antibiotics—F, G, and H—three dietary regimens—M, N, and O—and three physical therapies—U, V, and W. For each case of the illness, a doctor will prescribe exactly five of the treatments, in accordance with the following conditions: If two of the antibiotics are prescribed, the remaining antibiotic cannot be prescribed. There must be exactly one dietary regimen prescribed. If O is not prescribed, F cannot be prescribed. If W is prescribed, F cannot be prescribed. G cannot be prescribed if both N and U are prescribed. V cannot be prescribed unless both H and M are prescribed.
Question: If O is prescribed for a given case, which one of the following is a pair of treatments both of which must also be prescribed for that case?
(A) F, M (B) G, V (C) N, U (D) U, V (E) U, W

```
treatments = [F, G, H, M, N, O, U, V, W]
antibiotics = [F, G, H]
dietary_regimens = [M, N, O]
physical_therapies = [U, V, W]
prescribed = Function(treatments, bool)
Count([t:treatments], prescribed(t)) == 5
Count([a:antibiotics], prescribed(a)) <= 2
Count([d:dietary_regimens], prescribed(d)) == 1
Implies(Not(prescribed(O)), Not(prescribed(F)))
Implies(prescribed(W), Not(prescribed(F)))
Implies(And(prescribed(N), prescribed(U)), Not(prescribed(G)))
Implies(prescribed(V), And(prescribed(H), prescribed(M)))

solve(Implies(prescribed(O), And(prescribed(U), prescribed(V)))) # (A)
solve(Implies(prescribed(O), And(prescribed(G), prescribed(V)))) # (B)
solve(Implies(prescribed(O), And(prescribed(N), prescribed(U)))) # (C)
solve(Implies(prescribed(O), And(prescribed(U), prescribed(V)))) # (D)
solve(Implies(prescribed(O), And(prescribed(U), prescribed(W)))) # (E)
```

---

### CoT (annotated by our authors)

---

Nine different treatments are available for a certain illness: three antibiotics—F, G, and H—three dietary regimens—M, N, and O—and three physical therapies—U, V, and W ...... (omitted, see above)
Question: If O is prescribed for a given case, which one of the following is a pair of treatments both of which must also be prescribed for that case?
(A) F, M (B) G, V (C) N, U (D) U, V (E) U, W

```
Let's first analyze the conditions. We know there must be exactly one dietary regimen
prescribed. If O is a prescribed dietary regimen, then M and N must not be prescribed.
We know V cannot be prescribed unless both H and M are prescribed, so V cannot be
prescribed.
Let's consider option (A). M must not be prescribed. This option is False.
Let's consider option (B). V cannot be prescribed. This option is False.
Let's consider option (C). N must not be prescribed. This option is False.
Let's consider option (D). V cannot be prescribed. This option is False.
Let's consider option (E). Let's assume U is not prescribed. Because exactly 5 of the
treatments will be prescribed, they must be F, G, H, O, and W. In this way both F and
W are prescribed, which contradicts the condition that if W is prescribed, F cannot be
prescribed. So U must be prescribed. We can prescribe G, H, O, U, W. This option is
True.
So the answer is (E).
```

---

Figure 14: Prompt (excerpt) used for LSAT. Another example can be found in Figure 1. Several of our authors authored different CoT prompts, leading to similar performance (ranging from 20% to 22%).

**Prompts for BOARDGAMEQA**

---

**SATLM**

---

A few players are playing a boardgame. The current state of the game is as follows. The gecko has 13 friends, and hates Chris Ronaldo. And the rules of the game are as follows. Rule1: If the gecko has more than 8 friends, then the gecko does not proceed to the spot that is right after the spot of the bat. Rule2: Regarding the gecko, if it is a fan of Chris Ronaldo, then we can conclude that it does not proceed to the spot that is right after the spot of the bat. Rule3: If something does not proceed to the spot right after the bat, then it does not give a magnifier to the swordfish.

Q: Based on the game state and the rules and preferences, does the gecko give a magnifier to the swordfish?

```
# If the gecko has more than 8 friends, then the gecko does not proceed to the spot
that is right after the spot of the bat.
Implies(has_more_than_8_friends(gecko), Not(proceed_to_spot_right_after(gecko, bat)))
# Rule2: Regarding the gecko, if it is a fan of Chris Ronaldo, then we can conclude
that it does not proceed to the spot that is right after the spot of the bat.
Implies(is_fan_of_chris_ronaldo(gecko), Not(proceed_to_spot_right_after(gecko, bat)))
# Rule3: If something does not proceed to the spot right after the bat, then it does
not give a magnifier to the swordfish.
ForAll([x],  Implies(Not(proceed_to_spot_right_after(x,  bat)),  Not(give_magnifier(x,
swordfish))))

# The current state of the game is as follows.  The gecko has 13 friends, and hates
Chris Ronaldo.
# The gecko has 13 friends.
has_more_than_8_friends(gecko) == True
# The gecko hates Chris Ronaldo.
is_fan_of_chris_ronaldo(gecko) == False

# question: does the gecko give a magnifier to the swordfish?
solve(give_magnifier(gecko, swordfish))
```

---

**CoT** from Kazemi et al. (2023)

---

A few players are playing a boardgame. The current state of the game is as follows. The gecko has 13 friends, and hates Chris Ronaldo. And the rules of the game are as follows. Rule1: If the gecko has more than 8 friends, then the gecko does not proceed to the spot that is right after the spot of the bat. Rule2: Regarding the gecko, if it is a fan of Chris Ronaldo, then we can conclude that it does not proceed to the spot that is right after the spot of the bat. Rule3: If something does not proceed to the spot right after the bat, then it does not give a magnifier to the swordfish.

Q: Based on the game state and the rules and preferences, does the gecko give a magnifier to the swordfish?

A: We know the gecko has 13 friends, 13 is more than 8, and according to Rule1 "if the gecko has more than 8 friends, then the gecko does not proceed to the spot right after the bat", so we can conclude "the gecko does not proceed to the spot right after the bat". We know the gecko does not proceed to the spot right after the bat, and according to Rule3 "if something does not proceed to the spot right after the bat, then it doesn't give a magnifier to the swordfish", so we can conclude "the gecko does not give a magnifier to the swordfish". So the statement "the gecko gives a magnifier to the swordfish" is disproved. The answer is no.

Figure 15: Prompt (excerpt) used for BOARDGAMEQA.

**Prompts for CLUTRR**

| SATLM |
| --- |

Dorothy took her daughter Michelle and her mother Gabrielle car shopping.
Q: How is [Michelle] related to [Gabrielle]?
```
# [Dorothy] took her daughter [Michelle] and her mother [Gabrielle] car shopping.
relation(Dorothy, Michelle) = (mother, daughter)
relation(Dorothy, Gabrielle) = (daughter, mother)
# How is [Michelle] related to [Gabrielle]?
solve(relation(Michelle, Gabrielle))
```

Teresa and her brother Ellis were having a wonderful time at Disneyland. Ellis asked his grandmother, Molly, to read him a bedtime story. Molly read him Hansel & Gretel, which the boy always loved. Sandra is married to Thomas, the couple welcomed Teresa into the world.
Q: How is [Molly] related to [Sandra]?
```
# [Teresa] and her brother [Ellis] were having a wonderful time at Disneyland.
relation(Teresa, Ellis) = (sister, brother)
# [Ellis] asked his grandmother, [Molly], to read him a bedtime story.
relation(Ellis, Molly) = (grandson, grandmother)
# [Sandra] is married to Thomas, the couple welcomed [Teresa] into the world.
relation(Sandra, Teresa) = (mother, daughter)
# How is [Molly] related to [Sandra]?
solve (relation(Molly, Sandra))
```

| PROGLM from Lyu et al. (2023) |
| --- |

Dorothy took her daughter Michelle and her mother Gabrielle car shopping.
Q: How is [Michelle] related to [Gabrielle]?
```
# To answer this question, we write a program to answer the following subquestions:
# 1. How is [Michelle] related to [Dorothy]? (independent, support: "[Dorothy] took
her daughter [Michelle] and her mother [Gabrielle] car shopping.")
relation(Michelle, Dorothy) = daughter
# 2. How is [Dorothy] related to [Gabrielle]? (independent, support: "[Dorothy] took
her daughter [Michelle] and her mother [Gabrielle] car shopping.")
relation(Dorothy, Gabrielle) = daughter
# 3. Final answer: How is [Michelle] related to [Gabrielle]? (depends on 1, 2)
relation(Michelle, Gabrielle) = relation(Michelle, Dorothy) @ relation(Dorothy,
Gabrielle)
```

Teresa and her brother Ellis were having a wonderful time at Disneyland...... (omitted, see above)
Q: How is [Molly] related to [Sandra]?
```
# To answer this question, we write a program to answer the following subquestions:
# 1. How is [Molly] related to [Ellis]? (independent, support: "[Ellis] asked his
grandmother, [Molly], to read him a bedtime story.")
relation(Molly, Ellis) = grandmother
# 2. How is [Ellis] related to [Teresa]? (independent, support: "[Teresa] and her
brother [Ellis] were having a wonderful time at Disneyland.")
relation(Ellis, Teresa) = brother
# 3. How is [Teresa] related to [Sandra]? (independent, support: "[Sandra] is married
to Thomas, the couple welcomed [Teresa] into the world.")
relation(Teresa, Sandra) = daughter
# 4. Final answer: How is [Molly] related to [Sandra]? (depends on 1, 2, 3)
relation(Molly, Sandra) = relation(Molly, Ellis) @ relation(Ellis, Teresa) @
relation(Teresa, Sandra)
```

Figure 16: Prompt (excerpt) used for CLUTRR.

**Prompts for PROOFWRITER**

---

**SATLM**

---

Here are some facts and rules:
If someone visits the squirrel and the squirrel visits the rabbit then they are round. All round people are not
kind. If someone is round then they chase the rabbit. If someone is red and they chase the rabbit then they
visit the dog. If someone is red then they visit the squirrel. If someone visits the squirrel then the squirrel
visits the rabbit. the rabbit visits the dog.
the squirrel chases the bald eagle. the squirrel chases the rabbit. the dog sees the bald eagle. the bald eagle
does not chase the dog. the bald eagle is red. the squirrel is round. the rabbit does not see the dog. the rabbit
sees the bald eagle. the rabbit sees the squirrel. the dog does not see the rabbit. the rabbit does not visit the
bald eagle. the dog does not chase the bald eagle.
Q: The statement "The bald eagle visits the dog" is True or False?

```
ForAll([x], Implies(And(visit(x, squirrel), visit(squirrel, rabbit)), round(x)))
ForAll([x], Implies(round(x), Not(kind(x))))
ForAll([x], Implies(round(x), chase(x, rabbit)))
ForAll([x], Implies(And(red(x), chase(x, rabbit)), visit(x, dog)))
ForAll([x], Implies(red(x), visit(x, squirrel)))
ForAll([x], Implies(visit(x, squirrel), visit(squirrel, rabbit)))
chase(squirrel, rabbit)
see(dog, bald_eagle)
Not(chase(bald_eagle, dog))
red(bald_eagle)
round(squirrel)
Not(see(rabbit, dog))
see(rabbit, bald_eagle)
see(rabbit, squirrel)
Not(see(dog, rabbit))
Not(visit(rabbit, bald_eagle))
Not(chase(dog, bald_eagle))

solve(visit(bald_eagle, dog))
```

---

**PROGLM** adapted from Creswell et al. (2023)

---

Here are some facts and rules:
If someone visits the squirrel and the squirrel visits the rabbit then they are round...... (omitted, see above)
Q: The statement "The bald eagle visits the dog" is True or False?

```
# the bald eagle is red.
bald_eagle_is_red = True
# If someone is red then they visit the squirrel.
bald_eagle_visits_squirrel = bald_eagle_is_red
# If someone visits the squirrel then the squirrel visits the rabbit.
squirrel_visits_rabbit = bald_eagle_visits_squirrel
# If someone visits the squirrel and the squirrel visits the rabbit then they are round.
bald_eagle_is_round = bald_eagle_visits_squirrel and squirrel_visits_rabbit
# If someone is round then they chase the rabbit.
bald_eagle_chases_rabbit = bald_eagle_is_round
# If someone is red and they chase the rabbit then they visit the dog.
bald_eagle_visits_dog = bald_eagle_is_red and bald_eagle_chases_rabbit
# Question: The statement "The bald eagle visits the dog" is True or False?
return bald_eagle_visits_dog
```

---

Figure 17: Prompt (excerpt) used for PROOFWRITER.

**Prompts for COLOREDOBJECT**

| SATLM |
| --- |

Q: On the table, you see a bunch of objects arranged in a row: a purple paperclip, a pink stress ball, a brown keychain, a green scrunchiephone charger, a mauve fidget spinner, and a burgundy pen. What is the color of the object directly to the right of the stress ball?

```
# What is the color of the object directly to the right of the stress ball?

stress_ball = next(x:objects, name(x) == 'stress ball')

direct_right = next(x:objects, index(x) - index(stress_ball) == 1)

solve(color(direct_right))
```

| PROGLM from Gao et al. (2023) |
| --- |

Q: On the table, you see a bunch of objects arranged in a row: a purple paperclip, a pink stress ball, a brown keychain, a green scrunchiephone charger, a mauve fidget spinner, and a burgundy pen. What is the color of the object directly to the right of the stress ball?

```
# Find the index of the stress ball
stress_ball_idx = None
for i, object in enumerate(objects):
  if object[0] == 'stress ball':
    stress_ball_idx = i
    break

# Find the directly right object
direct_right = objects[i+1]

# Check the directly right object's color
direct_right_color = direct_right[1]
answer = direct_right_color

return answer
```

Figure 18: Prompt (excerpt) used for COLORED OBJECT.

**Prompts for STRUCTURED REGEX**

---



**SATLM**



---

Find the regex for the described patterns. Each regex r can be composed using sub-regexes r1, r2, r3, ...

Pattern:
Three strings separated by semicolons. The first string can either be 579 or 719, the second and third are composed by three digits or three lower case letters that can be followed by a lower case letter, a digit or a capital letter.

```
r = concat(r1,concat(<;>,concat(r2,concat(<;>,r2))))
r1 = or(<579>,<719>)
r2 = concat(or(r3,r4),optional(r5))
r3 = repeat(<num>,3)
r4 = repeat(<low>,3)
r5 = or(<low>,or(<num>,<cap>))
```

---



**PROGLM** from Gao et al. (2023)



---

Find the regex for the described patterns.

Pattern:
Three strings separated by semicolons. The first string can either be 579 or 719, the second and third are composed by three digits or three lower case letters that can be followed by a lower case letter, a digit or a capital letter.

Regex:
```
concat(or(<579>,<719>),concat(<;>,concat(concat(or(repeat(<num>,3),repeat(<low>,3)),
optional(or(<low>,or(<num>,<cap>)))),concat(<;>,concat(or(repeat(<num>,3),repeat(<low>,3)),
optional(or(<low>,or(<num>,<cap>)))))))))
```

---

Figure 19: Prompt (excerpt) used for STRUCTURED REGEX.

**Prompts for SATCotSolver**

---



**SATCotSolver for GSM**



---

Q: Jason had 20 lollipops. He gave Denny some lollipops. Now Jason has 12 lollipops. How many lollipops did Jason give to Denny?

```
jason_lollipops_initial = 20
lollipops_given = Variable()
jason_lollipops_after = 12
jason_lollipops_after = jason_lollipops_initial - lollipops_given
result = lollipops_given
solve(result)
# Solve the value of result
# We know jason_lollipops_after = jason_lollipops_initial - lollipops_given
# So lollipops_given = jason_lollipops_initial - jason_lollipops_after = 20 - 12 = 8
# result = lollipops_given = 8
# The answer is 8
```

---



**SATCotSolver for CLUTRR**



---

Dorothy took her daughter Michelle and her mother Gabrielle car shopping.
Q: How is [Michelle] related to [Gabrielle]?
```
# [Dorothy] took her daughter [Michelle] and her mother [Gabrielle] car shopping.
relation(Dorothy, Michelle) = (mother, daughter)
relation(Dorothy, Gabrielle) = (daughter, mother)
# How is [Michelle] related to [Gabrielle]?
solve(relation(Michelle, Gabrielle))
# Solve the problem
# Gabrielle's daughter is Dorothy, and Dorothy's daughter is Michelle, so Gabrielle's
granddaughter is Michelle.
# The answer is granddaughter.
```

Figure 20: Prompt (excerpt) used for SATCotSolver.

