# OpenReview forum: "SatLM: Satisfiability-Aided Language Models Using Declarative Prompting"
_NeurIPS.cc/2023/Conference — NeurIPS 2023 poster_

### Official Review · Reviewer_nffC · 2023-07-05

**Soundness:** 4 excellent
**Presentation:** 3 good
**Contribution:** 3 good
**Rating:** 7
**Confidence:** 4

**Summary:**

This paper aims at improving reasoning with large language models (LLMs) by prompting them in a way that they parse the problem into a language that is understandable by a SAT solver, and then employ an off-the-shelf SAT solver to solve the problem. Empirical results on multiple datasets show the benefit of the proposed approach.

**Strengths:**

* The idea of obviating the need for planning by employing a SAT solver is neat.
* Empirical results are strong.
* The approach works on various tasks/datasets.
* The analysis in Table 5 is quite interesting.


**Weaknesses:**

* I believe the benchmarks used in this work may slightly overestimate the performance of the proposed model and the approach may not work as well on more realistic use cases (see the limitations section for more detail)
* It is not clear to me how some bodies of work can fit within the parse-plan-execute framework described on Page 4. Could you comment on how the approaches such as [1, 2] that use LLMs as a tool within a reasoning algorithm can be described in this framework? How about decomposition-based approaches such as [3, 4]?
* [minor] The descriptions in Section 3 and, to some extent, Section 2 could be significantly shortened and some experimental details could be moved to the supplementary (e.g., decoding strategy, temperature, etc.). This makes room for a more elaborate description of different categories of related work (as opposed to putting everything into one category) and for moving some example failures from the appendix to the main text.

[1] LAMBADA: Backward Chaining for Automated Reasoning in Natural Language

[2] Selection-inference: Exploiting large language models for interpretable logical reasoning

[3] Decomposed prompting: A modular approach for solving complex tasks

[4] Least-to-most prompting enables complex reasoning in large language mod

**Questions:**

* Could you comment on the applicability of the approach on more realistic use cases (see limitations section for detail)?
* It has been shown in [1] that in the case of logical reasoning, dealing with “unknown” examples is difficult for CoT. How does SatLM deal with unknowns (e.g., for the ProofWriter dataset)? Could the “selective prediction” analysis be extended to deal with “unknown” labels? e.g., the UNSAT class.
* Any intuition why SatLM works sub-par to ProgLM on the GSM dataset, but significantly outperforms it on the GSM-SYS subset?

**Limitations:**

Many of the current LLM reasoning datasets (including some of the benchmarks in this work) have been created by first generating a puzzle in a formal language and then turning it into text either using templates or using human annotators. Therefore, the performance of approaches that translate back to a formal language (including the current work) may be overestimated on these benchmarks. Could the authors comment on the applicability of their approach to more realistic applications in the following two cases as examples:


1- Consider a logical reasoning puzzle described below:

Fiona assassinated the mayor. If somebody killed the mayor, they must go to prison. Should Fiona go to prison?

My guess is that if you translate to a formal language, you will get something like this:

assassinated(Fiona, mayor)

killed(X, mayor) -> go(X, prison)

go(Fiona, prison)?

which makes the solver not be able to produce the correct answer.


2- Consider the mathematical puzzle below:

Fiona has 10 apples, 15 dragon fruits, and 11 coconuts. How many tropical fruits does Fiona have?

---

> ### Author Rebuttal · Authors · 2023-08-08
>
> Thank you for your thoughtful comments and feedback.
>
> **Q1: It is not clear how the approaches such as [1, 2] that use LLMs as a tool within a reasoning algorithm and how the decomposition-based approaches such as [3, 4] can be fit into the parse-plan-execute framework.**
>
> A: At a high level, all the approaches [1,2,3,4] generally follow a CoT paradigm and rely on LLMs to perform parsing, planning, and execution. ([3] involves calling external APIs such as search in some settings). But unlike the standard streamlined CoT prompting that performs the parsing, planning, and executing with a single LLM call, these approaches are different in that 1) the plans are built iteratively by LLMs through multiple LLM calls, and 2) the execution of individual reasoning steps are performed using designated prompts.
>
> In particular, [1,2] iteratively builds the plans by prompting LLMs to select the relevant facts, and iteratively executes individual steps by prompting LLMs to make the inference (e.g., deduce new facts). [3,4] alternatively calls a decomposer prompt and specialized prompts for solving sub-problems. Overall, these approaches all use LLMs to generate the plans and also use LLMs to execute them.
>
>
> **Q2: Could you comment on the applicability of the approach on more realistic use cases?**
>
> A: Thanks for the question and detailed examples! Powerful LLMs can indeed generate consistent specifications to handle NL statements that involve non-trivial NL-to-formula parsing (required in the first case) and some commonsense reasoning (in the second case).
>
> We’ll now dive into each of the two use cases.
>
> > Fiona assassinated the mayor. If somebody kills the mayor, they must go to prison. Should Fiona go to prison?
>
> Yes, this kind of reasoning is in scope for our method. When parsing the NL into specifications, the generation of one formula is conditioned on previously generated formulas as opposed to being generated independently. We observe empirically that LLMs can recognize such flexibility and generate consistent specifications.
>
> Concretely, when giving our prompt used for ProofWriter, SatLM returns:
>
> ```
>     kill(fiona, mayor)
>     ForAll([x], Implies(kill(x, mayor), prison(x)))
>     solve(prison(fiona))
> ```
>
> > Fiona has 10 apples, 15 dragon fruits, and 11 coconuts. How many tropical fruits does Fiona have?
>
> Yes, we can solve this. When given our arithmetic reasoning prompt, SatLM returns:
> ```
>     apples = 10
>     dragon_fruits = 15
>     coconuts = 11
>     tropical_fruits = dragon_fruits + coconuts
>     result = tropical_fruits
>     solve(result)
> ```
>
> Powerful LLMs will be able to perform commonsense reasoning implicitly and output correct formulas. The LLM successfully ignores apples when generating the specification in this case.
>
> Lastly, we give some additional output examples from GSM that reflect how LLMs can perform non-trivial parsing that requires commonsense reasoning.
>
> - For the NL statement “a rectangle with an area of 160”, SatLM generates the constraint `length * height = 160`.
>
> - For the NL query “How much does she spend on food in the month of May?”, the SatLM produces code that assigns 31 to variable days_in_may (`days_in_may = 31`).
>
> - For the NL query “Farmer Brown has 60 animals on his farm, all either chickens or cows. He has twice as many chickens as cows. How many legs do the animals have, all together?”  SatLM generates the constraints `legs_chickens = animals_chickens * 2.  legs_cows = animals_cows * 4`.
>
> **Q3: It has been shown in [1] that in the case of logical reasoning, dealing with “unknown” examples is difficult for CoT. How does SatLM deal with unknowns (e.g., for the ProofWriter dataset)? Could the “selective prediction” analysis be extended to deal with “unknown” labels?**
>
> A: While we follow [2] and primarily focus on the “closed-world” setting for the ProofWriter dataset, SatLM can naturally handle the “unknowns”.
>
> Specifically, in the “open-world” setting, the value of a statement is unknown if we cannot prove the statement has to be true or has to be false (in other words, there exists an assignment that makes the statement evaluate as true as well as an assignment that makes the statement evaluate as false). This essentially indicates an “AMBIG” solution. Therefore, our SatLM can naturally deal with the “unknowns”, if we let SatLM output “unknown” when spotting an AMBIG solution instead of signaling an error.
>
> We note that the data points in the “closed-world” setting used in our work and [2] have ensured that there are no ambiguous solutions. As a result, the SatLM always makes a Yes/No prediction if successfully executed.
>
> **Q4: Any intuition why SatLM works sub-par to ProgLM on the GSM dataset, but significantly outperforms it on the GSM-SYS subset?**
>
> A: Intuitively, our approach is a particularly good fit for settings where the problem description is declarative—i.e., the NL description specifies what should be solved rather than how to solve it. Our approach works better on GSM-SYS because task descriptions have this property, which is difficult ground for ProgLM. In contrast, many problems in the rest of the GSM dataset contain more step-by-step instructions for many tasks.
>
> We’d also like to note that, while SatLM works less well than ProgLM on GSM dataset with greedy decoding, SatLM outperforms ProgLM with self-consistency decoding. By drawing multiple samples, SatLM can increase its coverage and achieve higher accuracy than ProgLM since its predictions are more accurate. (see detailed discussion in line 275 - 278 of the paper).
>
> ---
>
> [1] LAMBADA: Backward Chaining for Automated Reasoning in Natural Language
>
> [2] Selection-inference: Exploiting large language models for interpretable logical reasoning
>
> [3] Decomposed prompting: A modular approach for solving complex tasks
>
> [4] Least-to-most prompting enables complex reasoning in large language mod

---

> > ### Comment · Reviewer_nffC · 2023-08-15
> > **Reviewer Response**
> >
> > Thanks for your responses and for providing the model outputs for the examples I provided. The fact that both of them were effectively solved using your fewshots for other tasks is quite exciting. Accordingly, I have increased my score.
> >
> > One final recommendation: To strengthen the argument regarding the performance of the proposed model on datasets with incomplete information, I encourage the authors to consider providing results on one such dataset. One possibility is the no-conflict subset of the BoardgameQA dataset (the dataset was seemingly made public only recently, so my score will not be affected if the authors don't incorporate it).

---

> > > ### Author Response · Authors · 2023-08-15
> > >
> > > Thanks for your response and for bringing up the BoardgameQA dataset, which is relevant and interesting. We will look into it and consider adding this evaluation.

---

### Official Review · Reviewer_C7Hb · 2023-07-05

**Soundness:** 3 good
**Presentation:** 4 excellent
**Contribution:** 3 good
**Rating:** 5
**Confidence:** 3

**Summary:**

This paper propose SATLM, which aims to solve the problem of planning error when using CoT and ProgramLM. Specifically, SATLM use a LLM to translate natural language problems into formal the language that is accepted by a solver and let the solver to do planning as well as calculation. The results on several reasoning datasets show the effectiveness of this method.

**Strengths:**

1. The paper propose to translate a problem into a formal form and then directly call a SAT solver, which is a good idea. The experiment results in Table 1 shows the effectiveness of the method.
2. The structure of the paper is clear and the paper is easy to follow. Figure 1 is informative enough to decribe the proposed method.
3. I like the ablation studies in this paper, which makes it clear which parts are important to the performance gain.

**Weaknesses:**

1. My main concern is the method may only perform well on simpler tasks such as GSM (I think it's GSM8K?) or Proofwriter. The question in  Proofwriter is nearly in the format of formal reasoning, which only requires line-by-line translation, which in turn makes the accuracy to be as high as 99.7%. On more complicated tasks such as MathQA and MATH, I think (1) the problems might not be able to expressed in first-order logic and (2) even they can, it's not easy to do the translation.
2. There are other works that combine symbolic solver with LLMs in a similar or different way. The authors have mentioned some in the related work part. However, none of them is empirically compared in this paper, and only listing the difference from them is not enough.

**Questions:**

See weaknesses. I don't have any further questions.

**Limitations:**

The paper has discussion of limitations in section 6, which I think is enough.

---

> ### Author Rebuttal · Authors · 2023-08-08
>
> Thank you for your thoughtful comments and feedback.
>
> **Q1: Can more complicated tasks such as those in MathQA and MATH be expressed in first-order logic? If expressible, how intuitive would it be to translate more complicated tasks to first-order logic formulas?**
>
> A: Our work can handle most of the problems in MathQA since the operations in these problems are mainly equalities and arithmetic, which are supported in first-order logic theories. We believe most of the problems are intuitive to translate to FOL as well.
>
> We show some example outputs of SatLM (obtained using our arithmetic reasoning prompt) on data instances from MathQA below.
>
> > What quantity of water should be added to reduce 9 liters of 50% acidic liquid to 30% acidic liquid?
>
> ```
> liters_initial = 9
> percent_initial = 50
> percent_desired = 30
> water_added = Variable()
> liters_after = liters_initial + water_added
> percent_after = percent_initial * liters_initial / liters_after
> percent_after = percent_desired
> ```
>
> > From a pack of 52 cards, 1 card is drawn at random. What is the probability that a red king is drawn
>
> ```
> cards_total = 52
> red_kings = 2
> probability = red_kings / cards_total
> ```
>
>
> In contrast, for the MATH dataset, we agree with the reviewer that it might be difficult to encode some of the problems as a FOL formula, in particular those problems that involve reasoning on graphs and geometry, for instance:
>
> > [Figure] Beginning at point A in the diagram, Dora selects one of the four possible directions with equal probability. Each time she comes to an intersection, she again randomly selects one of the possible directions. What is the probability that, in her first four steps, she will walk completely around the gray square? Express your answer as a common fraction.
>
>
> Many of the problems in MATH also require symbolic computations, such as:
>
> > The equation x^2 + 2x = i has two complex solutions. Determine the product of their real parts.
>
> While doing symbolic computations is currently not supported in SMT solvers, we could potentially handle these problems in the same framework using a richer logic and more powerful solver like mathematica. Furthermore, we would like to point out that the main purpose of the MATH dataset is to evaluate the ability of machine learning models doing complicated reasoning, while we show how to offload complicated reasoning to symbolic solvers. Our framework shows the capability of LLMs in the presence of strong solvers; we believe that solvers will always be the most reliable way to handle problems like this at the frontiers of LLMs’ capabilities.
>
>
> **Q2: There are other works that combine symbolic solver with LLMs in a similar or different way. The authors have mentioned some in the related work part. However, none of them is empirically compared in this paper, and only listing the difference from them is not enough.**
>
> A: Actually, we do **directly compare** against a line of program-aided LMs [1,2,3] in the evaluation, all of which are techniques that equip LLMs with symbolic executors (program interpreters). In particular, we use the implementation from [1] for the ProgLM results on GSM and COLOR, use the implementation from [2] for the results on CLUTRR, and use our own implementation for other tasks where existing implementations are not available (see Appendix B for more details of implementation). These program-aided systems set the state-of-the-art performance on these tasks at the time of our experiments. We would be happy to consider evaluating more techniques as suggested by the reviewer.
>
> Furthermore, in **Appendix D**, we also include an apples-to-apples comparison between our work and concurrent work [4] that focuses on using symbolic solvers for solving math problems. Our approach achieves better performance than theirs on  arithmetic reasoning (see Table 10 in Appendix D) and is more widely applicable across a diverse range of tasks beyond arithmetic reasoning.
>
> [1] PAL: Program-Aided Language Models. Gao et al., 2023.
>
> [2] Faithful Chain-of-thought Reasoning. Lyu et al., 2023.
>
> [3] Program of Thoughts Prompting: Disentangling Computation from Reasoning for Numerical Reasoning Tasks. Chen et al., 2022.
>
> [4] Solving Math Word Problems by Combining Language Models with Symbolic Solvers. He-Yueya et al., 2023.

---

> > ### Comment · Reviewer_C7Hb · 2023-08-10
> >
> > Thank you for your reply. I still find your description of baseline methods a little bit confusing. For examples, you are comparing with different baselines for different datasets, though I understand that there might not be ready-made prompts. Also I would recommend moving some key points in Appendix B to the main text to avoid confusion.

---

> > > ### Author Response · Authors · 2023-08-10
> > >
> > > Thanks for your reply and suggestions.
> > >
> > > > For examples, you are comparing with different baselines for different datasets, though I understand that there might not be ready-made prompts.
> > >
> > > Yes. The baseline, ProgLM, covers implementations of symbolic executor-aided approaches from several papers [1,2], and each of them only includes ready-made prompts for a subset of the dataset studied in our paper. We also note that because they all follow exactly the same paradigm that produces a python program and employs a python interpreter to execute the program, we unify them together in a framework called ProgLM.
> > >
> > > We will make this clearer in any future version. We will also move the comparison with concurrent symbolic solver-aided work to the main text if page limits allow.

---

### Official Review · Reviewer_Zc4x · 2023-07-06

**Soundness:** 3 good
**Presentation:** 3 good
**Contribution:** 2 fair
**Rating:** 6
**Confidence:** 4

**Summary:**

This paper presents a framework to augment LLMs with symbolic solvers to compensate known flaws in LLMs' reasoning (e.g., planning and arithmetic). The main novelty of this paper is that LLMs is prompted to generate declarative specifications (rather programs) so that off-the-shelf automatic theorem provers like Z3 can be deployed to solve the symbolic problems.

**Strengths:**

Augmenting LLMs with symbolic solvers is a super valuable direction to explore. Upgrading the symbolic solver from a programming language interpreter to an automatic theorem prover surely will benefit LLMs further. This paper is well motivated, nicely structured and has compelling experiments.

**Weaknesses:**

The main weakness to me is that SatLM does not differentiate itself enough from the previous [Faithful CoT paper](https://github.com/veronica320/Faithful-COT) as a framework. Besides parsing sub-problems into a programming language like Python, Faithful CoT can also put those problems into Planning Domain Definition Language (PDDL), which is, in a sense, specification.

**Questions:**

- line 101, [Maieutic Prompting](https://arxiv.org/abs/2205.11822) appears to be relevant here.
- line 118, I know that authors decided to use Python rather than a specialized DSL here, but it would be great if the authors can elaborate on the syntax of the specification.
- Table 1, it would be interesting to see results on more advanced models like gpt-3.5-turbo and gpt-4, as the performance of [Faithful CoT](https://github.com/veronica320/Faithful-COT) increased quite a bit on these models.

Minor:
- Figure 1, 'an NL input'
- line 318, did you miss a citation here?

**Limitations:**

The authors have adequately addressed the limitations.

---

> ### Author Rebuttal · Authors · 2023-08-08
>
> Thank you for your thoughtful comments and feedback.
>
> **Q1: How does SatLM differentiate itself enough from the previous Faithful CoT paper as a framework?**
>
> A: SatLM uses SAT specifications that can encode a wide range of reasoning problems spanning arithmetic reasoning, logical reasoning, and symbolic reasoning, which are further solved with a unified solver. In contrast, the PPDL specification and planner used in Faithful-CoT or LLM+P is specifically designed for planning problems.
>
> Furthermore, we’d like to point out that for the math world problems (GSM8K, specifically) and logical inference problems (CLUTTER, specifically) used in both our work and Faithful-CoT, Faithful-CoT uses an imperative specification (python program), which is fundamentally different from ours.
>
> **Q2: How does SatLM relate to Maieutic Prompting?**
>
> A: Maieutic prompting differs in two fundamental ways. First, it elicits statements from the language model, including both correct and incorrect statements, and tries to identify a logically-consistent subset. Our approach only generates correct statements. Second, maieutic prompting uses soft logical relations between statements derived from an entailment model, making the execution very different from using a Z3-based solver.
>
> **Q3: Please elaborate on the syntax of the specification.**
>
> A: We include the answer to Q5 to reviewer Gm6N in the following.
>
> > The specification used in our work largely follows and simplifies the syntax for specifying constraints used in [z3 python](https://z3prover.github.io/api/html/namespacez3py.html).
> >
> > We list concrete example statements as follows. See Figure 1, Figure 2 and Appendix G for more examples.
> >
> > ```
> > # declare a variable
> > x = Variable()
> > # declare enum set
> > People = [Alice, Bob]
> > Cities = [Austin, Boston]
> > Food = [Apple, Banana]
> > # declare function
> > visit = Function(People, Cities)
> > eats = Function(People, Food)
> >
> > # logic
> > visit(Alice) != visit(Bob)
> > # quantifier
> > ForAll(x: People, Implies(visit(x) == Austin, eats(x) == Banana))
> > ```
> >
> > We note that these formulas are close to the actual python-code formulas used by z3 python but are slightly modified to be more amenable to prompting, so we need a postprocessing step to form the actual Z3 input. We implemented a simple parser that transforms these formulas into actual specifications used by z3 python via string transformation (using regexes). For example, we transform `ForAll(x: People, Implies(visit(x, Austin), eats(x, Banana)))` into `x = Variable(People) ForAll([x], Implies(visit(x) == Austin, eats(x) == Banana))`.
>
>
> **Q4: What happens if you run the evaluations on more advanced models such as GPT-3.5-turbo or GPT-4?**
>
> A: We include the results of SatLM and baselines (CoT, ProgLM) on gpt-3.5-turbo-0613 below. The results suggest the same trend as the results on other LLMs. SatLM outperforms both CoT and ProgLM across all datasets except on GSM.
>
> |        | GSMSys | GSM  | Algebra | LSAT | CLUTRR | Proof |
> |--------|--------|------|---------|------|--------|-------|
> | CoT    | 44.8   | 74.4 | 29.3    | 23.9 | 41.2   | 82.3  |
> | ProgLM | 51.2   | 77.9 | 62.6    | -    | 45.9   | 76.4  |
> | SatLM  | 63.4   | 76.4 | 77.9    | 30.0 | 50.5   | 98.8  |

---

> > ### Comment · Reviewer_Zc4x · 2023-08-19
> > **Thanks a lot for the clarification**
> >
> > I appreciate the detailed response from the authors.
> >
> > > A: SatLM uses SAT specifications that can encode a wide range of reasoning problems spanning arithmetic reasoning, logical reasoning, and symbolic reasoning, which are further solved with a unified solver. In contrast, the PPDL specification and planner used in Faithful-CoT or LLM+P is specifically designed for planning problems.
> >
> > I admit that there are subtle differences between planning and SMT solving, but I don't think they are drastically different in the benchmarking scenarios in this paper. As a specification language, both PPDL and SMT-LIB (I assume you are calling Z3 through this interface) are heavily underutilized here.
> >
> > I understand that the authors want to emphasise that SatLM uses a unified solver (for the symbolic tasks), but I still yet to see the clear advantage of using just one solver against various symbolic solvers including programming language interpreters (e.g., Python) and PPDL solvers (which again can be a SMT solver like Z3).

---

> > > ### Author Response · Authors · 2023-08-19
> > >
> > > Thanks for your response. We’d like to note two things:
> > >
> > > 1. While this work uses few-shot prompting, we believe that future work can exploit various types of fine-tuning (e.g., reinforcement learning with outcome-based feedback) to improve performance at the conversion to our declarative specifications. In this case, having a unified format across problems enables a single fine-tuned model to more effectively multi-task.
> > >
> > > 2. We do want to note that in addition to unifying the solver, our work particularly emphasizes the use of declarative specifications, whereas Faithful CoT uses a mix of imperative and declarative specifications across problems. In particular, on arithmetic and logical reasoning tasks, where our work uses declarative specification as opposed to the imperative specification used in Faithful CoT, our choice of specifications achieves significant performance gain compared to Faithful CoT across LLMs (see Table 1, Table 6, Table 10, and the initial author response).

---

> > > > ### Comment · Reviewer_Zc4x · 2023-08-21
> > > > **Thanks again for the further clarification**
> > > >
> > > > I greatly appreciate the authors' effort of resolving my concerns. I have increased my score slightly.

---

### Official Review · Reviewer_p8AX · 2023-07-07

**Soundness:** 3 good
**Presentation:** 3 good
**Contribution:** 2 fair
**Rating:** 5
**Confidence:** 4

**Summary:**

This work looks at using an LLM to generate a declarative task specification from a natural language specification for reasoning tasks and leverage an automated SAT solver to solve the problem. They showcase that SATLM performs better than using Chain of thought or ProgramLM (which converts natural language into python programs) in various reasoning tasks.

**Strengths:**

The paper is well-written, the experiments are thorough.

**Weaknesses:**

The main contribution of this work seems to be about using LLMs as a semantic parser (for mostly short-context problems) which essentially maps a problem from NL specification to a formal specification (SAT problem specification here). This is a well-known problem in NLP for which LLMs (particularly LLMs which are trained on text and code) are known to perform comparably to the supervised neural semantic parsers [1,2]. The results of this work essentially indicate whether LLMs can perform NL-to-SAT semantic parsing given that once the parsing is done correctly, the SAT solver solves it. This makes the comparison between SATLM and ProgLM/CoT a bit unfair as the core ability that has to be compared is the parsing ability and not the planning/execution ability and these are intertwined in the latter. Comparisons between SATLM and other semantic parsers (other LLMs as done in Section 4.5 and other neural task-specific semantic parsers) for NL-to-SAT semantic parsing has to be highlighted in my opinion.

[1] Shin, R., & Van Durme, B. (2022, July). Few-Shot Semantic Parsing with Language Models Trained on Code. In Proceedings of the 2022 Conference of the North American Chapter of the Association for Computational Linguistics: Human Language Technologies (pp. 5417-5425).
[2] Zhuo, T. Y., Li, Z., Huang, Y., Li, Y. F., Wang, W., Haffari, G., & Shiri, F. (2023). On robustness of prompt-based semantic parsing with large pre-trained language model: An empirical study on codex. arXiv preprint arXiv:2301.12868.

**Questions:**

1. Is the LLM producing anything that is semantically new and not present in the natural language specification?

2. Shouldn’t COT also be tested with text models as opposed to code models given that code models are optimized for more formal representations?

3. Just for clarification, in Table 5, does “Answered” for ProgLM mean that the program was executable without runtime errors (but it need not return the right answer)?

**Limitations:**

I have provided the limitations of the work in the weaknesses section. The authors do address important limitations such as problems being less suitable to declarative prompting, limitations of SAT solvers when dealing with complex formulas and potential future work on re-prompting the LLM with the SAT solver feedback.

---

> ### Author Rebuttal · Authors · 2023-08-08
>
> Thank you for your thoughtful comments and feedback.
>
> > The main contribution of this work seems to be about using LLMs as a semantic parser.
>
> We disagree a bit with this interpretation. Our main point is that, for these types of reasoning problems, it makes more sense to decompose the problem into two parts: (1) Parsing the query into a declarative specification; and (2) Solving it using a theorem prover. This is in contrast to the ProgLM approach that generates a program to solve the end-to-end task. As such, we view the contribution of this paper as a reasoning framework, powered by LLMs and SAT solvers, as opposed to an LLM-based semantic parser.
>
> We note that both the SatLM framework and ProgLM framework follow a semantic parsing paradigm, where LLMs parse an NL problem into a formal specification that is executed by a symbolic executor. But the use of declarative specifications allows SatLM to more accurately parse the descriptions than ProgLMs. Furthermore, we note that most prior work in semantic parsing handles single-sentence queries, not long and complex problems like those in our datasets (e.g., each SAT specification for GSM8k involves more than 7 constraints on average).
>
> > The comparison between SatLM and ProgLM/CoT a bit unfair as the core ability that has to be compared is the parsing ability and not the planning/execution ability
>
> The comparison between SatLM and ProgLM essentially compares LLMs’ parsing abilities in terms of producing SAT-style specifications and program-style specifications. We note as both SatLM and ProgLM use symbolic executors to execute the parsed specifications, which ensures correct execution with respect to the parse outputs, so the improvements of SatLM over ProgLM can be attributed to the fact that using declarative specifications leads to more accurate parsing compared to using imperative specifications.
>
> We agree that the solver we use (Z3) does have a greater planning capacity than the Python interpreter used in much of the past ProgLM work. However, we view that as an inherent strength of the framework proposed here.
>
> > Comparisons between SatLM and other semantic parsers for NL-to-SAT semantic parsing has to be highlighted in my opinion.
>
> This work uses LLMs to parse NL into SAT specifications via few-shot prompting, similar to [1]. We are not claiming novelty on the idea of building a semantic parser this way. To the best of our knowledge, there aren't any other existing NL-to-SAT semantic parsers that we can compare to, but if the reviewer is aware of tools that can be used here, we would be happy to evaluate them in our framework.
>
> [1] Few-Shot Semantic Parsing with Language Models Trained on Code.  Shin & Van Durme. 2022.
>
> ---
>
> **Q1: Is the LLM producing anything that is semantically new and not present in the natural language specification?**
>
> A: Interpreting the natural language to produce a SAT specification does require making inferences. Here are some examples:
>
> - For the NL statement “a rectangle with an area of 160”, SatLM generates the constraint `length * height = 160`.
>
> - For the NL query “How much does she spend on food in the month of May?”, SatLM produces code that assigns 31 to variable days_in_may (`days_in_may = 31`).
>
> - For the NL query “Farmer Brown has 60 animals on his farm, all either chickens or cows. He has twice as many chickens as cows. How many legs do the animals have, all together?”, SatLM generates the constraints `legs_chickens = animals_chickens * 2.  legs_cows = animals_cows * 4`.
>
>
> **Q2: Shouldn’t COT also be tested with text models as opposed to code models given that code models are optimized for more formal representations?**
>
> A: On these tasks, even with text-focused models, CoT usually lags ProgLM by a large margin (see Lyu et al., 2023; Gao et al., 2023). We have also included the results of CoT on text-davinci-003 below. As seen in the table, CoT generally underperforms both ProgLM and SatLM on text-davinci-003, which agrees with past work (Lyu et al., 2023; Gao et al., 2023).
>
> We have additionally tested on chat-focused models (gpt-3.5-turbo), which shows the same trend. Please see Q4 in the response to Reviewer Zc4x for details.
>
>
>
> |        | GSMSys | GSM | LSAT | CLUTRR | Proof |
> |--------|--------|------|------|--------|-------|
> | CoT    | 42.8 | 66.5 | 21.7 | 34.5 | 83.5 |
> | ProgLM | 51.2 | 71.7 | - | 41.2 | 83.7 |
> | SatLM  | 63.4 | 70.3 | 30.4 | 58.2 | 99.7 |
>
>
> **Q3: In Table 5, does “Answered” for ProgLM mean that the program was executable without runtime errors (but it need not return the right answer)?**
>
> A: Yes, “Answered” means the program interpreter or SAT solver successfully executes the program or SAT problem, respectively, and returns a not necessarily correct answer. We see SatLM makes less predictions but has substantially higher selective accuracy.

---

> > ### Comment · Reviewer_p8AX · 2023-08-21
> >
> > Thank you for the clarifications and results of COT on text-based models. I have updated the score slightly.

---

### Official Review · Reviewer_Gm6N · 2023-07-07

**Soundness:** 2 fair
**Presentation:** 3 good
**Contribution:** 2 fair
**Rating:** 5
**Confidence:** 3

**Summary:**

This paper describes a new approach to solving NL reasoning tasks using large language models (LLMs). Specifically, the key idea is to combine LLMs with SAT solving, where LLMs are only used in the first parsing step. The authors of the paper call this approach satisfiability-aided language modeling (SATLM). The authors also point out that by SAT, they mean all kinds of tools for automated reasoning, including traditional SAT solvers, SMT, and first-order theorem provers. The idea is that the declarative specification that is generated as output from prompting is closer and more accurate compared to more imperative problem formulation. The work is evaluated on several data sets with standard prompting, chain-of-thought (COT) prompting, and executor-augmented LLMs (PROGLM).

**Strengths:**


- The paper is relatively easy to read and strikes a good balance between formality and accessibility.

- The main idea of the paper is clear, and it seems to be a new direction of how to encode NL tasks using LLMs.

- The paper contains relevant related work and contains relevant benchmarks





**Weaknesses:**

- There is an interesting discussion on how the proposed approach can catch errors better than PROGLM (see e.g., the paragraph starting on line 166). This is good, but there is no discussion on how this kind of error can be handled. How easy is it to debug such errors? How can a user generate a fix or solution if such an error occurs?

- The results are not always that easy to interpret. For instance, in table 2, if an answer is incorrect, does it then include both incorrect because of parse error, or just incorrect because of the SAT solver generating the wrong answer? This is much clearer in table 5.

- The level of detail of how the approach has actually been implemented could be improved. More specifically, how are the different tools chained together, are all steps automated etc.


**Questions:**

How do we know that SatLM generated a correct SAT problem? Can we get guarantees? If not, how can we debug the program?

On lines 127-127, you explain that you use Python instead of a declarative DSL. Please provide more details on what the target language looks like for the SAT problem specification. In what ways does it differ for traditional SAT solvers, SMT, and first-order theorem provers?

It would be useful to get a deeper understanding of the benchmarks. In the cases where there is a pure SAT problem (true or false), what is then the distribution of TRUE and FALSE answers? If the distribution is 50/50, would not then a random generator with uniform distribution perform better than many of the solutions that are compared?

**Limitations:**

There is a clear paragraph on the limitations of the approach at the end of the paper.

---

> ### Author Rebuttal · Authors · 2023-08-08
>
> Thank you for your thoughtful comments and feedback.
>
>
> **Q1: While SatLM can capture more types of errors (specifically, UNSAT and AMBIG) than PROGLM, there is no discussion on how this kind of error can be handled. How easy is it to debug such errors? How can a user generate a fix or solution if such an error occurs?**
>
> A: The reviewer brings up very interesting questions about user interaction with SatLM. Our current prototype just produces UNSAT or AMBIG, but we can think of several directions for providing more feedback in these cases:
>
> - UNSAT errors:  We can show the user an UNSAT core, which is a (small) subset of the constraints that are unsatisfiable. This can help the user to more quickly identify any problems.
>
> - AMBIG errors: We could potentially use abductive reasoning to suggest additional constraints. In particular, abduction can generate additional constraints under which the constraint system would have a unique solution.
>
> This work opens the door to explore exciting questions like these. We believe there is potential both to allow more effective intervention by human users (e.g., by recognizing errors in the specification) as well as using iterative prompting of LLMs to improve the specification once the error has been localized. However, a full treatment of this topic is outside the scope of this paper.
>
>
> **Q2: In Table 2, if an answer is incorrect, does it then include both incorrect because of parse error, or just incorrect because of the SAT solver generating the wrong answer?**
>
> A: The accuracy in Table 2 refers to the accuracy of the final answer caused by either parsing errors or solving errors made by LLM solvers (CoTSolver and NoSolver). We will make this clearer in any future version.
>
>
> **Q3: How do we know that SatLM generated a correct SAT problem? Can we get guarantees? If not, how can we debug the program?**
>
> A: To be 100% sure that SatLM generated the correct SAT problem, the user would need to inspect the generated output. We believe that inspecting the SAT problem is much easier than inspecting a program that produces the solution because it is declarative and much higher-level
>
> **Q4: How are the different tools chained together, are all steps automated etc.**
>
> A: First, we prompt an LLM using declarative prompting (described in Section 3.2) to output the SAT specification (see Section 4.1 for details of the decoding method). Next, the output (SAT specification) is post-processed into actual z3 python code (see the response to the next question for details). Lastly, we execute the z3 python code to obtain the final answer. These steps are chained together automatically without needing manual intervention. We can clarify this more in any future version!
>
> **Q5: Please provide more details on what the target language looks like for the SAT problem specification. In what ways does it differ from traditional SAT solvers, SMT, and first-order theorem provers?**
>
> A: The specification used in our work largely follows and simplifies the syntax for specifying constraints used in [z3 python](https://z3prover.github.io/api/html/namespacez3py.html).
>
> We list concrete example statements as follows. See Figure 1, Figure 2 and Appendix G for more examples.
>
> ```
> # declare a variable
> x = Variable()
> # declare enum set
> People = [Alice, Bob]
> Cities = [Austin, Boston]
> Food = [Apple, Banana]
> # declare function
> visit = Function(People, Cities)
> eats = Function(People, Food)
>
> # logic
> visit(Alice) != visit(Bob)
> # quantifier
> ForAll(x: People, Implies(visit(x) == Austin, eats(x) == Banana))
> ```
>
> We note that these formulas are close to the actual python-code formulas used by z3 python but are slightly modified to be more amenable to prompting, so we need a postprocessing step to form the actual Z3 input. We implemented a simple parser that transforms these formulas into actual specifications used by z3 python via string transformation (using regexes). For example, we transform `ForAll(x: People, Implies(visit(x, Austin), eats(x, Banana)))` into `x = Variable(People) ForAll([x], Implies(visit(x) == Austin, eats(x) == Banana))`.
>
>
> **Q6: In the cases where there is a pure SAT problem (true or false), what is then the distribution of TRUE and FALSE answers? If the distribution is 50/50, would not then a random generator with uniform distribution perform better than many of the solutions that are compared?**
>
> A: We’d like to clarify that the SAT problem used in our work is not restricted to “pure SAT” problems but can refer to problems that require checking the satisfiability of formulas in formal logic. Most problems studied in our paper are not binary classification tasks where the final answer is to assess whether a statement is true or false.
>
> Only the ProofWriter dataset is a binary classification dataset.  As the labels are balanced for the ProofWriter datasets, all the solutions perform much better than random guessing (50%).

---

> > ### Comment · Reviewer_Gm6N · 2023-08-16
> >
> > Thanks for the clarifications and the concrete examples. When you iterate the manuscript, I think it would be great to include more details on how the tools are chained together and to be more specific about the target language.

---

> > > ### Author Response · Authors · 2023-08-16
> > >
> > > Thanks for your response and suggestions. We will integrate these details into the main text in any future versions.

---

### Decision · Program_Chairs · 2023-09-21

**Decision:**

Accept (poster)

**Comment:**

The paper presents a new approach to introduce logic reasoning into LLMs by using the LLM to translate the task into a formal notation that can be fed into an off-the-shelf solver. The approach shows significant advantages compared to state-of-the art prompting approaches. The rebuttal showed some additional impressive results against some queries posed by a reviewer which illustrated the ways in which the approach can leverage the domain knowledge in the LLM to do a faithful translation to logic. As reviewers pointed out, the idea of translating problems from natural language into a format that a solver can reason about has been explored in other contexts, but the paper and the rebuttal make a strong case for the power and generality of this particular approach.